# Molecular Heterogeneity in Early-Onset Colorectal Cancer: Pathway-Specific Insights in High-Risk Populations

**DOI:** 10.3390/cancers17081325

**Published:** 2025-04-15

**Authors:** Cecilia Monge, Brigette Waldrup, Francisco G. Carranza, Enrique Velazquez-Villarreal

**Affiliations:** 1Center for Cancer Research, National Cancer Institute, Bethesda, MD 20892, USA; 2Department of Integrative Translational Sciences, Beckman Research Institute, City of Hope, Duarte, CA 91010, USA; 3City of Hope Comprehensive Cancer Center, Duarte, CA 91010, USA

**Keywords:** early-onset colorectal cancer, cancer disparities, genetic mutations, precision medicine, WNT pathway, TGF-beta pathway, RTK/RAS pathway

## Abstract

Early-onset colorectal cancer (EOCRC), diagnosed before the age of 50, is increasing rapidly, especially among Hispanic/Latino individuals, who face higher rates of both diagnosis and death from the disease. However, little is known about the genetic differences that may contribute to these disparities. This study aims to identify specific genetic changes in key cancer-related pathways in EOCRC among Hispanic/Latino and non-Hispanic White individuals. By comparing mutation patterns in these groups, the researchers hope to uncover biological differences that may influence cancer development and outcomes. Understanding these differences could lead to more personalized and effective treatment strategies, particularly for high-risk populations. Ultimately, this research may help improve care and reduce the burden of colorectal cancer in underrepresented communities.

## 1. Introduction

Colorectal cancer (CRC) remains a critical global health issue, ranking as the third most prevalent cancer and the second leading cause of cancer-related deaths worldwide [1]. While advancements in screening and early detection have led to stable or declining CRC incidence in high-income countries, the incidence of early-onset colorectal cancer (EOCRC), diagnosed before the age of 50, has been increasing at an alarming rate [2,3,4,5]. This rising trend is particularly concerning, as EOCRC is often associated with a more aggressive disease course and worse prognosis than late-onset colorectal cancer (LOCRC) [6]. Since standard CRC screening guidelines typically recommend routine screening beginning at age 50, many EOCRC cases are detected at later stages, posing a significant public health challenge [7].

Among racial and ethnic groups, the H/L population in the United States has experienced the highest increase in EOCRC incidence and mortality [8,9]. Despite this growing burden, the molecular drivers of EOCRC in H/L individuals remain poorly characterized with limited sample sizes [10,11,12]. The limited inclusion of H/L patients in large-scale genomic studies has hindered efforts to identify ethnicity-specific oncogenic mechanisms that may contribute to disparities in CRC incidence, progression, and clinical outcomes [13,14]. Addressing these disparities necessitates a deeper investigation into the molecular landscape of EOCRC in high-risk populations, particularly in relation to key oncogenic pathways.

Distinct molecular features have been observed in EOCRC, including higher microsatellite instability (MSI), increased tumor mutation burden, and elevated PD-L1 expression compared to LOCRC [3,13,14]. Additionally, LINE-1 hypomethylation has been proposed as a unique biomarker of EOCRC [15]. Comparative genomic analyses have highlighted significant differences in key oncogenic mutations between EOCRC and LOCRC, particularly in genes such as TP53, SMAD4, BRAF, NOTCH1, CTNNB1, APC, and KRAS [13,16]. However, few studies have specifically examined how these molecular alterations differ between H/L and non-Hispanic White (NHW) EOCRC patients, further underscoring the need for population-specific analyses.

Three critical signaling pathways implicated in CRC development and progression—the WNT, TGF-beta, and RTK/RAS pathways—play central roles in tumor initiation, invasion, and therapeutic resistance. The WNT pathway regulates β-catenin activation, which drives CRC progression and is commonly altered through mutations in APC, CTNNB1, and RNF43 [17,18]. Notably, WNT pathway activation is a hallmark of CRC, with mutations in APC observed in over 80% of cases [19]. Studies suggest that EOCRC may exhibit unique WNT pathway alterations compared to LOCRC, with some analyses indicating a lower prevalence of WNT mutations in EOCRC [16], while others report elevated β-catenin activation [20,21]. A recent study demonstrated that EOCRC patients with WNT pathway alterations exhibited improved survival compared to those without, but this effect was observed primarily in NHW populations [10].

Similarly, the TGF-beta signaling pathway plays a crucial role in CRC progression, promoting epithelial-to-mesenchymal transition (EMT), immune evasion, and metastasis [22]. TGF-beta pathway dysregulation is frequently observed in CRC, particularly through mutations in SMAD4 and BMPR1A, which have been associated with tumor aggressiveness and poor prognosis [23,24]. Genome-wide association studies (GWASs) have identified 16 genes within the TGF-beta pathway that are significantly associated with EOCRC [25]. Among H/L patients, EOCRC-specific alterations in BMP7 and BMPR1A have been identified, suggesting population-specific differences in TGF-beta signaling [10].

The RTK/RAS pathway, which regulates cell proliferation, survival, and differentiation, is frequently altered in CRC. Mutations in KRAS, NRAS, and BRAF are among the most common genetic alterations in CRC, with KRAS mutations occurring in approximately 40% of cases [26]. These mutations confer resistance to anti-EGFR therapies, a major challenge in CRC treatment [27]. EOCRC patients exhibit higher frequencies of KRAS and NRAS mutations compared to LOCRC [7]. However, recent studies suggest that RTK/RAS pathway alterations may be less prevalent in EOCRC compared to LOCRC in H/L patients, with an enrichment in mutations in CBL, NF1, and MAPK3 instead [10].

With the rising incidence of EOCRC among the H/L population and the limited understanding of ethnicity-specific oncogenic mechanisms, this study aims to explore the molecular heterogeneity in EOCRC in high-risk populations using a robust sample size. This approach seeks to address the historical underrepresentation of this group in clinical and genomic research, which is essential for advancing new technologies aimed at uncovering the molecular underpinnings of this cancer [28,29]. By examining pathway-specific alterations in WNT, TGF-beta, and RTK/RAS signaling, we aim to identify key molecular differences between EOCRC cases in H/L and NHW individuals [30,31]. Furthermore, we assess the clinical implications of these alterations to support the development of precision medicine strategies tailored to underrepresented populations. Gaining insight into the distinct molecular landscape of EOCRC in H/L patients is crucial for designing targeted therapeutic interventions and improving clinical outcomes in this high-risk group.

## 2. Materials and Methods

### 2.1. Clinical and Genomic Data

This study leveraged clinical and genomic data from 20 publicly available CRC datasets accessible through the cBioPortal database. The analyzed datasets encompassed colorectal adenocarcinoma, colon adenocarcinoma, and rectal adenocarcinoma, along with data from the GENIE BPC CRC v2.0-public dataset. To maintain a focus on primary tumor cases, datasets specifically examining metastatic CRC were excluded. Patient selection followed predefined inclusion criteria, requiring identification as Hispanic or Latino, Spanish, NOS; Hispanic, NOS; Latino, NOS; or individuals with a Mexican or Spanish surname. Additional filtering parameters ensured that only primary tumor cases were included, limiting selection to colorectal, colon, and rectal adenocarcinomas with confirmed adenocarcinoma and NOS histology, allowing only one sample per patient. All analyses were performed using R statistical software (version 4.4.1).

After applying these criteria, four datasets—TCGA PanCancer Atlas, MSK Nat Commun 2022, MSK-CHORD, and GENIE BPC CRC—met this study’s requirements, resulting in a cohort of 302 H/L patients, including 138 EOCRC and 164 LOCRC cases. Similarly, 3110 NHW patients (897 EOCRC and 2213 LOCRC) were identified using identical inclusion criteria (Table 1 and Table 2). Age at diagnosis was extracted from GENIE database clinical records.

WNT, TGF-beta, and RTK/RAS pathway alterations were identified using established criteria, and the patients were categorized into EOCRC, diagnosed before age 50, and LOCRC, diagnosed at age 50 or older. Further stratification was performed based on ethnicity (H/L vs. NHW) and the presence or absence of alterations in the aimed oncogenic pathways. Molecular alterations in the WNT, TGF-beta, and RTK/RAS pathways were identified based on gene-level data annotated in cBioPortal. Mutation types included frameshift deletions, frameshift insertions, missense mutations, nonsense mutations, splice site mutations, and translation start site mutations. MSI status and MSI scores were extracted from clinical annotations; the samples were categorized as microsatellite stable, microsatellite instability, indeterminate, do not report, and unavailable based on dataset-specific classifications.

To define pathway-specific gene sets, we curated gene lists for the WNT, TGF-β, and RTK/RAS signaling pathways using the established literature on colorectal cancer genomics [32,33]. Only genes with well-documented involvement in colorectal cancer tumorigenesis and pathway regulation were included. Mutation data were extracted at the gene level from cBioPortal, and we included somatic alterations classified as nonsynonymous, including missense, nonsense, frameshift insertions/deletions, splice site mutations, and translation start site mutations. Genes were analyzed individually and as part of their respective pathways to distinguish between alterations in central versus peripheral components of signaling cascades. Pathway alteration burden was defined as the presence of at least one qualifying mutation in any pathway member gene. This approach allowed for a balanced interpretation of both mutation frequency and potential biological impact in the context of pathway disruption.

Table 3 details the distribution of these pathway alterations among the EOCRC and LOCRC patients within the H/L cohort, illustrating their prevalence across different age groups. Table 4 expands upon this analysis by comparing EOCRC cases between the H/L and NHW patients, allowing for a comparative evaluation of pathway-specific molecular differences between ethnic groups.

### 2.2. Statistical Analysis

Chi-square tests were employed to compare mutation frequencies across groups, assessing the independence of categorical variables and examining associations between age, ethnicity, and pathway alterations. To enhance the analysis, tumor samples were further categorized by anatomical location (colon vs. rectal adenocarcinoma), providing a more detailed evaluation of how the tumor site interacts with ethnicity and pathway alterations.

To assess gene-level mutation frequencies and pathway-specific differences among the H/L patients with colorectal cancer, we conducted a gene-by-gene analysis of somatic mutations across the WNT, TGF-β, and RTK/RAS signaling pathways. For each gene, the presence or absence of nonsynonymous mutations—including missense, nonsense, frameshift, splice site, and translation start site variants—was calculated as a proportion of the total cases in the early-onset colorectal cancer (EOCRC) and late-onset colorectal cancer (LOCRC) groups. Comparisons of mutation frequencies between the EOCRC and LOCRC cases were conducted using chi-square tests or Fisher’s exact tests, depending on the distribution and sample size of the data. This pathway-informed, gene-level approach allows for the identification of both broad pathway trends and specific driver gene alterations. The full list of genes and their corresponding mutation frequencies by age group are provided in Appendix A, which offer a comprehensive view of how pathway disruptions differ by age within the H/L population.

Kaplan–Meier survival analysis was conducted to determine the prognostic impact of WNT, TGF-beta, and RTK/RAS pathway alterations on overall survival in different patient subgroups. Survival curves were generated to visualize variations in survival probability over time, comparing patients with and without these pathway alterations. The log-rank test was used to identify statistically significant differences between survival distributions, while median survival times and 95% confidence intervals (CIs) were calculated to ensure precise survival estimates. By leveraging large-scale genomic data, survival analyses, and subgroup comparisons, this study provides an in-depth evaluation of pathway-specific disruptions in EOCRC and LOCRC, particularly among H/L patients.

## 3. Results

Using data from four cBioPortal projects that included ethnicity information, we established two cohorts: 302 H/L patients and 3110 NHW patients. Within the H/L cohort, 27.5% of the males and 18.2% of the females were diagnosed with EOCRC before the age of 50, while 30.8% of the males and 23.5% of the females were diagnosed at age 50 or older (LOCRC). Comparatively, the NHW cohort had lower proportions of EOCRC cases, with 16.2% of the males and 12.7% of the females diagnosed before 50, whereas 38.9% of the males and 32.3% of the females were diagnosed with LOCRC (Table 1).

The gender distribution was relatively consistent across the cohorts, with the H/L group comprising 55% males and 45% females, and the NHW group consisting of 56% males and 44% females. Regarding cancer stage at diagnosis, 32.5% of the H/L patients were diagnosed at stages 0, I, II, or III, while 43.7% had stage IV disease. In the NHW cohort, 31.0% of the patients were diagnosed at earlier stages (0–III), and 36.4% were diagnosed at stage IV. A notable proportion of cases had missing or unreported staging data, with 23.8% of the H/L patients and 32.6% of the NHW patients recorded as “NA” for stage at diagnosis.

Ethnicity classification within the H/L cohort showed that the majority (89.4%) identified as Spanish NOS, Hispanic NOS, or Latino NOS. Additionally, 9.3% of the patients were classified as Mexican (including Chicano), while 1.3% identified as Other Spanish/Hispanic or were categorized based on a Spanish surname. In contrast, all of the NHW patients (100%) were classified as NHW, ensuring a clear distinction between the two groups for comparative analysis.

These demographic and clinical characteristics highlight notable differences in EOCRC incidence and disease stage at diagnosis between H/L and NHW populations, emphasizing potential disparities that could influence clinical outcomes.

A comparative analysis of clinical characteristics revealed significant differences between EOCRC and LOCRC in the H/L patients, as well as between the EOCRC H/L and EOCRC NHW patients (Table 2). The median age at diagnosis for the EOCRC H/L patients was 41 years (IQR: 36–46), which was notably younger than the median age of 61 years (IQR: 55–69) for the LOCRC H/L patients (*p* < 0.05). Similarly, the EOCRC H/L patients had a significantly lower median age at diagnosis compared to the EOCRC NHW patients, whose median age was 42 years (IQR: 38–47) (*p* < 0.05).

Mutation burden analysis showed that the EOCRC H/L patients had a median mutation count of seven, slightly lower than the median of eight observed in the LOCRC H/L patients; however, this difference was not statistically significant (*p* > 0.05). Likewise, the EOCRC NHW patients exhibited a median mutation count of seven, which was similar to that of the EOCRC H/L patients, with no significant difference between the two groups (*p* > 0.05).

Pathway-specific analyses revealed significant differences in genetic alterations within the WNT, TGF-beta, and RTK/RAS pathways when comparing EOCRC and LOCRC in the H/L cohort. Notably, mutations in NF1 (11.6% vs. 3.7%, *p* < 0.05), CBL (5.8% vs. 1.2%, *p* < 0.05), and BRAF (5.1% vs. 18.3%, *p* < 0.05) demonstrated significant variation between the EOCRC and LOCRC cases. While BRAF mutations were significantly more prevalent in the LOCRC cases, NF1 and CBL mutations were more frequently observed in the EOCRC patients. Similarly, BMPR1A mutations were detected more often in the EOCRC patients (5.1%) compared to the LOCRC cases (1.2%), though this difference was not statistically significant (*p* > 0.05).

A comparison between the EOCRC H/L and NHW CRC patients further revealed ethnic-specific molecular differences. RNF43 (12.3% vs. 6.7%, *p* < 0.05) from the WNT pathway, BMPR1A (5.1% vs. 1.8%, *p* < 0.05) from the TGF-beta pathway, and MAPK3 (3.6% vs. 0.7%, *p* < 0.05), CBL (5.8% vs. 1.4%, *p* < 0.05), and NF1 (11.6% vs. 6.1%, *p* < 0.05) from the RTK/RAS pathway were significantly more prevalent in the EOCRC H/L patients compared to the NHW patients. However, the frequency of BRAF mutations did not significantly differ between the two ethnic groups (5.1% vs. 7.5%, *p* > 0.05). These findings highlight potential ethnicity-specific genomic differences that may play a role in CRC pathogenesis, warranting further investigation into their functional and clinical significance.

The variations in WNT, TGF-beta, and RTK/RAS pathway alterations observed between EOCRC and LOCRC, as well as between the H/L and NHW EOCRC patients, emphasize the necessity for further research into their biological and clinical significance. A deeper understanding of these molecular differences could contribute to the development of targeted therapeutic approaches designed to reduce disparities in CRC outcomes across diverse ethnic groups.

Our analysis of genetic alterations in H/L individuals with EOCRC and LOCRC revealed no statistically significant differences in the prevalence of TGF-beta and WNT pathway alterations (Table 3). TGF-beta pathway mutations were identified in 34.1% of the EOCRC cases and 33.5% of the LOCRC cases, with identical absence rates of 65.9% and 66.5%, respectively (*p* = 1). Similarly, WNT pathway alterations were highly frequent in both groups, affecting 89.9% of the EOCRC patients and 84.1% of the LORCR patients, though the difference was not statistically significant (*p* = 0.1979). Notably, the absence of WNT pathway alterations was slightly higher in the LOCRC patients (15.9%) compared to the EOCRC patients (10.1%).

In contrast, significant differences were observed in RTK/RAS pathway alterations between the EOCRC and LOCRC patients. RTK/RAS mutations were detected in 66.7% of the EOCRC cases, whereas they were more prevalent in the LOCRC cases at 79.3% (*p* = 0.01922). Conversely, the proportion of patients without RTK/RAS alterations was higher in EOCRC (33.3%) compared to LOCRC (20.7%). These results indicate that while TGF-beta and WNT pathway mutations are consistently present in both age groups, RTK/RAS pathway alterations may have a greater influence on tumor development in LOCRC within the H/L population.

Additional research is needed to better understand the biological significance of RTK/RAS pathway variations between EOCRC and LOCRC patients. Moreover, examining how these pathways interact with other molecular drivers could offer valuable insights into CRC progression and identify potential therapeutic targets for this high-risk population.

Our analysis of genetic alterations in EOCRC among the H/L and NHW individuals revealed significant differences in the frequency of TGF-beta pathway mutations, while no notable differences were observed in WNT and RTK/RAS pathway alterations (Table 4). TGF-beta pathway mutations were significantly more prevalent in the EOCRC H/L patients (34.1%) compared to the EOCRC NHW patients (25.5%) (*p* = 0.04489). In contrast, the NHW individuals had a higher proportion of cases without TGF-beta pathway alterations (74.5%) compared to the H/L individuals (65.9%).

Alterations in the WNT pathway were commonly observed in both EOCRC groups, affecting 89.9% of the H/L patients and 84.1% of the NHW patients. However, this difference did not reach statistical significance (*p* = 0.101). Conversely, the proportion of patients without WNT pathway alterations was slightly higher among the NHW individuals (15.9%) compared to the H/L individuals (10.1%).

RTK/RAS pathway alterations were detected in 66.7% of the EOCRC H/L patients and 65.2% of the EOCRC NHW patients, showing no significant difference between the two groups (*p* = 0.8126). Likewise, the proportion of patients without RTK/RAS pathway alterations remained similar, with 34.8% in the NHW group and 33.3% in the H/L group.

The results indicate that TGF-beta pathway alterations occur at a significantly higher frequency in EOCRC among H/L individuals, whereas WNT and RTK/RAS pathway mutations show no substantial differences between the two populations.

Survival outcomes for EOCRC in the H/L patients, analyzed using Kaplan–Meier and stratified by the presence or absence of TGF-beta pathway alterations (Figure 1), showed no statistically significant difference in overall survival (*p* = 0.74). The survival trajectories of both groups were largely overlapping, indicating that TGF-beta pathway mutations may have a minimal impact on survival outcomes within this cohort. Although the patients with TGF-beta alterations showed a slightly lower survival probability during the initial months of follow-up, their survival curve eventually aligned with that of the patients without alterations. The broad confidence intervals, especially at later time points, suggest variability in survival outcomes and the potential influence of additional molecular or clinical factors not accounted for in this analysis. These results imply that TGF-beta pathway alterations alone may not be strong prognostic markers in EOCRC H/L patients. Further research incorporating larger cohorts and multi-omics analyses is needed to explore whether interactions with other molecular pathways contribute to survival disparities in this high-risk population.

The analysis of overall survival in the EOCRC H/L patients, based on the presence or absence of WNT pathway alterations (Figure 1), showed no statistically significant difference (*p* = 0.31). Although the patients with WNT pathway alterations had slightly lower survival probabilities in the early months of follow-up, the survival curves later converged, indicating no clear link between WNT pathway mutations and long-term survival. Notably, these results differ from findings in the NHW EOCRC patients, where WNT alterations were associated with better survival outcomes. The absence of statistical significance in the H/L cohort suggests that the prognostic role of WNT pathway mutations may vary across ethnic groups. Additional studies with larger sample sizes and functional investigations are needed to determine whether interactions with other oncogenic pathways contribute to survival differences in EOCRC among H/L patients.

Unlike the findings for TGF-beta and WNT pathway alterations, the survival analysis for the EOCRC H/L patients based on RTK/RAS pathway alterations (Figure 1) indicated a possible trend toward worse survival outcomes in those with RTK/RAS mutations (*p* = 0.87). The patients with RTK/RAS alterations experienced an early decline in survival probability, maintaining lower survival rates throughout the follow-up period compared to those without such alterations. Conversely, the patients without RTK/RAS pathway mutations exhibited a more stable survival probability over time. Although the difference did not reach statistical significance, the observed trend suggests that RTK/RAS mutations could play a role in survival disparities among EOCRC patients in the H/L population.

Survival analysis using the Kaplan–Meier method for the EOCRC NHW patients, categorized by the presence or absence of TGF-beta pathway alterations, showed no statistically significant difference in overall survival (*p* = 0.53) (Appendix A). The survival curves of both groups remained nearly identical, with minimal divergence, indicating that TGF-beta pathway alterations may not be a strong prognostic factor in NHW EOCRC. Although the patients with TGF-beta alterations exhibited slightly lower survival probabilities in the early follow-up period, the curves eventually converged, suggesting no consistent survival trend. The wide confidence intervals and variability in survival estimates emphasize the need for further research with larger cohorts. Future investigations should examine whether TGF-beta alterations interact with other molecular or clinical factors that may impact disease progression and treatment response in NHW EOCRC patients.

Conversely, survival analysis using the Kaplan–Meier method for the NHW EOCRC patients, categorized by the presence or absence of WNT pathway alterations, revealed a statistically significant association with improved overall survival (*p* = 0.0015) (Appendix A). The patients with WNT pathway mutations (red curve) consistently maintained higher survival probabilities across the follow-up period compared to those without mutations (blue curve), indicating a potential protective role of WNT alterations in this population. The clear separation between survival curves further supports a survival advantage for patients harboring WNT pathway mutations.

Kaplan–Meier survival analysis of the NHW EOCRC patients, stratified by RTK/RAS pathway alterations, showed no statistically significant difference in overall survival (*p* = 0.18) (Appendix A). While the RTK/RAS-altered group (red curve) displayed a slight trend toward reduced survival compared to those without alterations (blue curve), the overlapping curves and wide confidence intervals indicated substantial variability in survival estimates.

To evaluate potential age-related variations, the frequencies of gene alterations associated with the WNT, TGF-beta, and RTK/RAS pathways were examined in the H/L patients diagnosed with EOCRC and LOCRC (Appendix A). The findings revealed distinct differences in the prevalence of specific pathway alterations between the two groups.

In the WNT pathway, APC mutations were more prevalent among the EOCRC patients (80.4%) than the LOCRC patients (70.7%), though the difference was not statistically significant (*p* = 0.07022). Other key WNT pathway genes, such as AXIN1, AXIN2, and RNF43, displayed similar mutation frequencies between the two groups, with no significant variation. Likewise, no substantial differences in mutation rates were identified within the TGF-beta pathway between the EOCRC and LOCRC patients. The most frequently altered gene in this pathway, SMAD4, had nearly identical mutation rates in both groups (14.5% in EOCRC vs. 14.6% in LOCRC, *p* = 1.0). Although BMPR1A mutations were slightly more common in the EOCRC patients (5.1%) compared to the LOCRC patients (1.2%), this difference did not reach statistical significance (*p* = 0.08853).

Within the RTK/RAS pathway, notable differences in gene alterations were observed between the EOCRC and LOCRC patients. NF1 mutations were significantly more frequent in the EOCRC cases (11.6%) compared to the LOCRC cases (3.7%) (*p* = 0.01547). Likewise, CBL mutations were more prevalent in the EOCRC patients (5.8%) than in the LOCRC patients (1.2%) (*p* = 0.04765). Additionally, MAP2K1 mutations were found exclusively in the EOCRC patients (3.6%) and were completely absent in the LOCRC cases (*p* = 0.01914). On the other hand, BRAF mutations were significantly more common in the LOCRC patients (18.3%) compared to the EOCRC patients (5.1%, *p* = 0.0009188), suggesting an age-related pattern in RTK/RAS pathway alterations.

A comparative analysis of EOCRC between the H/L and NHW patients revealed notable disparities in the mutation rates of key pathway genes (Appendix A). Within the WNT pathway, RNF43 mutations were significantly more prevalent among the EOCRC H/L patients (12.3%) compared to their NHW counterparts (6.7%, *p* = 0.02985). This suggests the possibility of ethnicity-specific variations in tumor suppressor gene alterations.

Within the TGF-beta pathway, BMPR1A mutations were notably more prevalent in the EOCRC H/L patients (5.1%) than in their NHW counterparts (1.8%, *p* = 0.04443). In contrast, other key genes in this pathway, including TGFBR2 and SMAD4, did not exhibit statistically significant differences between the two groups. This finding implies that BMPR1A mutations may have a more prominent role in the tumorigenesis of EOCRC among H/L individuals.

In the RTK/RAS pathway, several genes displayed significantly higher mutation frequencies in the EOCRC H/L patients compared to their NHW counterparts. Specifically, MAPK3 mutations were more prevalent in the EOCRC H/L patients (3.6%) than in the EOCRC NHW patients (0.7%, *p* = 0.006833). Similarly, NF1 mutations occurred at a significantly higher rate in the H/L patients (11.6%) versus the NHW patients (6.1%, *p* = 0.02907). Additionally, CBL mutations were nearly four times more common in the EOCRC H/L patients (5.8%) compared to the NHW patients (1.4%, *p* = 0.002302). These findings suggest potential ethnic variations in RTK/RAS pathway alterations, which may play a role in CRC development and influence therapeutic responses.

On the other hand, the mutation frequencies for KRAS, NRAS, and BRAF did not differ significantly between the EOCRC H/L and NHW patients. This indicates that although certain RTK/RAS pathway genes exhibit ethnic-specific enrichment, the primary driver mutations appear to be consistent across both populations.

Analysis of mutation types within the WNT signaling pathway revealed notable differences in the nature of gene alterations across both age and ethnic groups (Appendix A). In the Hispanic/Latino EOCRC cases, RNF43 mutations were predominantly frameshift deletions (48.0%), whereas APC mutations were largely nonsense mutations (51.4%), followed by frameshift deletions (23.6%). In contrast, AXIN1 and GSK3B mutations in this group were exclusively missense mutations. Among the H/L LOCRC patients, RNF43 mutations also frequently presented as frameshift deletions (51.3%), but a broader range of alterations, including missense (28.2%) and frameshift insertions (10.3%), was observed. AXIN2 mutations in this group were primarily disruptive, with over half being frameshift deletions (54.5%). For the non-Hispanic White EOCRC patients, RNF43 alterations showed an even higher prevalence of frameshift deletions (56.3%), while APC mutations were heavily enriched for nonsense mutations (51.3%). AXIN2 alterations were more evenly distributed among missense (32.7%), frameshift deletion (36.7%), and insertion-type mutations. In NHW LOCRC, a similarly high rate of APC nonsense mutations was observed (54.8%), and RNF43 mutations followed a consistent pattern of dominant frameshift deletions (59.4%).

The distribution of mutation types across the TGF-beta pathway genes demonstrated distinct patterns between the early-onset and late-onset colorectal cancer cases, as well as between the H/L and NHW populations (Appendix A). Among the H/L EOCRC patients, most TGF-beta pathway alterations were missense mutations, particularly in TGFBR1 (85.7%), BMPR1A (85.7%), SMAD2 (70.0%), and SMAD3 (87.5%). TGFBR2 mutations in this group were a mix of frameshift deletions (41.7%) and missense mutations (58.3%), while SMAD4 exhibited a broader distribution, including frameshift deletions (9.1%), insertions (9.1%), and nonsense mutations (13.6%). In contrast, the H/L LOCRC cases showed a marked increase in truncating mutations, particularly nonsense mutations, across key genes: SMAD2 (75.0%), SMAD3 (72.7%), SMAD4 (74.1%), and TGFBR1 (66.7%). TGFBR2 maintained a balanced split between frameshift deletions (46.2%) and nonsense mutations (53.8%), indicating substantial loss-of-function events in older patients. Among the NHW EOCRC cases, the mutation patterns were more diverse. SMAD2, SMAD3, and SMAD4 alterations were dominated by missense mutations (55.8–75.8%), though truncating mutations (nonsense and frameshift) were also present in 11.7–26.1% of the SMAD4 and SMAD2 cases. TGFBR2 mutations included frameshift deletions (33.3%), missense mutations (52.4%), and low-frequency insertion and nonsense events. Interestingly, mutations in BMPR1A showed a substantial presence of nonsense (35.3%) and missense (52.9%) variants, and TGFB3 and other BMP family members were exclusively altered via missense mutations in EOCRC. In the NHW LOCRC samples, a similar pattern persisted, with a dominant prevalence of missense mutations across TGF-beta pathway genes. Notable exceptions included TGFBR2 (37.7% frameshift deletions; 6.0% nonsense), SMAD4 (10.6% nonsense), and BMPR1A (20.4% nonsense). Across the BMP subfamily, including BMP4, BMP6, and BMP7, mutations remained strictly missense, while truncating alterations were enriched in BMPR2 and BMP10.

Mutation profiling of RTK/RAS pathway genes revealed both shared and divergent alteration patterns across the age and ethnicity groups (Appendix A). Among the H/L EOCRC patients, most alterations were missense mutations, particularly in key oncogenes such as BRAF, KRAS, NRAS, MAPK3, and NTRK family members, where 100% of mutations were missense. Certain regulators, including NF1 and RASA1, exhibited greater mutational diversity, with NF1 showing frameshift deletions (21.2%), insertions (12.1%), and nonsense (9.1%) mutations. EGFR and MET also displayed structural disruptions, including splice site mutations in 20% and 33.3% of the cases, respectively. In the H/L LOCRC cases, nonsense mutations and structural variants were more frequent. NF1 showed a notable increase in nonsense (37.5%) and splice site (37.5%) alterations. Genes like ALK, ARAF, and MET harbored high rates of frameshift deletions (22.2%, 66.7%, and 28.6%, respectively). CBL, KIT, RET, and ROS1 mutations in this group were exclusively nonsense or splice site, indicating likely inactivation. Additionally, complex alteration types emerged in FLT3 and RASA1, which showed mixed insertion, deletion, and splice site patterns. For the non-Hispanic White EOCRC patients, the mutations remained predominantly missense in canonical RTK/RAS genes (KRAS, NRAS, BRAF, EGFR, ERBBs), although a subset of genes—such as NF1, RASA1, SOS1, and ALK—exhibited more heterogeneous alterations, including frameshifts (up to 13.2%), nonsense, and splice site mutations. For example, NF1 had frameshift and nonsense variants in over 25% of cases. CBL mutations in NHW EOCRC were particularly diverse, including missense (66.7%), nonsense (13.3%), and insertions (6.7%). In NHW LOCRC, the mutation landscape was even more diverse, with several genes showing multi-class alterations, including MAP2K1, FGFRs, ALK, ERRFI1, ROS1, and RIT1. Though missense mutations remained dominant (especially in KRAS, NRAS, and BRAF), truncating alterations were more frequent in NF1 (13.5% nonsense), CBL (10.9% nonsense), and MET (11.4% nonsense). Genes like SOS1 and RASA1 also had significant proportions of frameshift deletions (22.8% and 18.6%, respectively), suggesting the inactivation of negative regulators in this pathway.

To explore the potential influence of complex molecular hallmarks on pathway-specific alterations, we evaluated microsatellite instability (MSI) status across the age and ethnicity groups (Appendix A). Among the H/L EOCRC patients, the majority of tumors were microsatellite stable (MSS, 70.3%), while only 8.0% were MSI-high, and 15.9% had unavailable MSI data. In contrast, MSI-high status was more frequent in H/L LOCRC (14.0%), despite a similarly high MSS rate (76.8%). A similar pattern was observed in the non-Hispanic White cohort, with NHW EOCRC cases showing 83.8% MSS and 8.2% MSI-high tumors, compared to the NHW LOCRC patients, where 12.7% were MSI-high and 72.7% MSS. The proportion of indeterminate or missing MSI data was notably higher in NHW LOCRC (14.6%) than in other groups. The MSI scores, a continuous measurement of instability burden, were lowest in the NHW EOCRC patients (median: 0.33; mean: 2.98), and highest in the NHW LOCRC patients (median: 0.51; mean: 4.85), suggesting increased MSI burden with age. Among the H/L patients, the median MSI scores were modestly higher in EOCRC (0.62) than in LOCRC (0.455), though mean values were slightly lower (3.80 vs. 4.65), indicating variability across individual tumors.

## 4. Discussion

CRC remains a significant public health challenge, with EOCRC exhibiting an alarming rise in incidence, particularly among H/L individuals. Despite this growing burden, the molecular mechanisms underlying EOCRC disparities remain poorly understood. This study provides a comprehensive analysis of WNT, TGF-beta, and RTK/RAS pathway alterations in EOCRC, revealing key differences between H/L and NHW patients. These findings contribute to the growing evidence that molecular heterogeneity plays a critical role in CRC pathogenesis and may inform precision medicine strategies tailored to underrepresented populations.

Our results highlight significant ethnic-specific differences in pathway alterations. Notably, RNF43, BMPR1A, MAPK3, NF1, and CBL mutations were significantly more prevalent in the EOCRC H/L patients compared to their NHW counterparts. RNF43 mutations, a critical regulator of WNT signaling, were observed in 12.3% of the EOCRC H/L patients compared to 6.7% of the NHW patients (*p* < 0.05), suggesting potential differences in WNT pathway dysregulation across ethnic groups. Similarly, BMPR1A mutations, implicated in TGF-beta signaling, were significantly enriched in the EOCRC H/L patients (5.1% vs. 1.8%, *p* < 0.05), reinforcing prior findings that alterations in this pathway may contribute to CRC progression in H/L individuals [5].

Within the RTK/RAS pathway, MAPK3 (3.6% vs. 0.7%, *p* < 0.01), NF1 (11.6% vs. 6.1%, *p* < 0.05), and CBL (5.8% vs. 1.4%, *p* < 0.01) mutations were significantly more frequent in the EOCRC H/L patients than in the NHW patients. The enrichment of NF1 mutations in the EOCRC H/L patients is particularly intriguing, as NF1 loss has been associated with resistance to targeted therapies in CRC and other malignancies [6,7]. These findings suggest that ethnic-specific RTK/RAS pathway alterations may influence tumor behavior and treatment response in EOCRC, emphasizing the need for further studies evaluating the functional impact of these mutations.

Comparisons between EOCRC and LOCRC within the H/L cohort revealed key distinctions in molecular alterations. While WNT and TGF-beta pathway mutations occurred at comparable frequencies across the age groups, RTK/RAS pathway alterations exhibited significant variation. NF1 (11.6% vs. 3.7%, *p* < 0.05), CBL (5.8% vs. 1.2%, *p* < 0.05), and MAP2K1 (3.6% vs. 0.0%, *p* < 0.05) mutations were significantly more prevalent in the EOCRC H/L patients, suggesting a potential role for RTK/RAS dysregulation in EOCRC. Conversely, BRAF mutations were markedly higher in the LOCRC patients (18.3% vs. 5.1%, *p* < 0.001), consistent with previous reports linking BRAF alterations to LOCRC and poor prognosis [8].

These findings support the notion that EOCRC and LOCRC may represent distinct molecular subtypes with unique oncogenic drivers. While NF1 and CBL mutations may contribute to tumor initiation in EOCRC, BRAF-driven signaling appears to play a more prominent role in LOCRC. The functional implications of these differences warrant further investigation, particularly in the context of targeted therapy development for EOCRC patients.

Kaplan–Meier survival analysis revealed significant differences in the impact of pathway alterations across the ethnic groups. In the EOCRC NHW patients, WNT pathway alterations were associated with improved survival (*p* = 0.0015), suggesting a potential protective role for WNT signaling mutations in this population. This aligns with prior studies indicating that APC mutations, a hallmark of WNT pathway activation, may be associated with better clinical outcomes in CRC [9]. However, this survival advantage was not observed in the EOCRC H/L patients, where WNT pathway mutations had no significant impact on overall survival (*p* = 0.31). These findings highlight potential ethnic-specific differences in the prognostic relevance of WNT signaling alterations, emphasizing the need for further investigation.

TGF-beta pathway alterations, in contrast, did not significantly impact survival outcomes in either the EOCRC H/L (*p* = 0.74) or NHW (*p* = 0.53) patients. The absence of a strong prognostic association suggests that TGF-beta mutations alone may not drive survival differences in EOCRC, though their interactions with other molecular pathways remain an important area of study.

RTK/RAS pathway alterations demonstrated a trend toward poorer survival outcomes in the EOCRC H/L patients (*p* = 0.087), though this did not reach statistical significance. This finding suggests that RTK/RAS pathway mutations may contribute to disease progression in EOCRC, potentially influencing treatment response. Interestingly, the EOCRC NHW patients with RTK/RAS alterations also exhibited a non-significant trend toward reduced survival (*p* = 0.18), reinforcing the need for larger cohort studies to validate these associations.

The molecular differences identified in this study have significant implications for precision medicine approaches in CRC. The high prevalence of RTK/RAS pathway mutations in the EOCRC H/L patients suggests that therapies targeting this pathway, such as MEK inhibitors, may be particularly relevant for this population. However, the presence of NF1 mutations in 11.6% of the EOCRC H/L patients raises concerns regarding potential resistance to RAS/MAPK-targeted therapies, highlighting the need for biomarker-driven treatment strategies [13,14].

The differential impact of WNT pathway alterations on survival outcomes between the NHW and H/L EOCRC patients suggests that ethnicity-specific molecular interactions may influence tumor progression. Future clinical trials should incorporate ethnic diversity in patient recruitment to better assess treatment responses in H/L CRC patients. Additionally, integrating multi-omics approaches—such as transcriptomic and proteomic profiling—will provide deeper insights into the functional consequences of pathway alterations in EOCRC.

The results indicate that certain RTK/RAS pathway alterations, including mutations in NF1, CBL, and MAP2K1, may have a greater impact on EOCRC, while BRAF mutations appear to be more prevalent in LOCRC. In contrast, the lack of significant variation in WNT and TGF-beta pathway alterations between the EOCRC and LOCRC onset cases suggests that these pathways may drive CRC progression independently of age at diagnosis. Additional research is required to investigate the functional consequences of these molecular differences and their potential applications in precision oncology approaches for H/L CRC patients.

While WNT and TGF-beta pathway alterations showed some ethnicity-specific variations, the most striking differences were observed in RTK/RAS pathway genes, especially NF1, CBL, and MAPK3. The significantly higher frequency of these mutations in EOCRC H/L patients highlights the necessity for further research into their functional impact on tumor development, resistance to therapy, and personalized treatment strategies for H/L CRC patients. Future investigations should focus on understanding how these molecular variations affect disease progression and response to targeted therapies across diverse racial and ethnic populations.

Our analysis of mutation types across the WNT, TGF-β, and RTK/RAS pathways revealed striking age- and ancestry-associated differences in the molecular mechanisms underlying colorectal cancer. In the WNT pathway, early-onset tumors in both the Hispanic/Latino and non-Hispanic White patients were characterized by a predominance of truncating mutations in APC and RNF43, particularly frameshift deletions and nonsense mutations, suggesting early disruption of WNT signaling as a common event in EOCRC. However, subtle ethnic differences were observed—such as a higher proportion of RNF43 frameshift deletions in NHW EOCRC and a broader mutation spectrum in AXIN2 among the H/L LOCRC cases—highlighting possible population-specific vulnerabilities. In the TGF-β pathway, mutation types varied substantially by both age and ethnicity. H/L EOCRC tumors showed a dominance of missense mutations in core signaling genes (TGFBR1, SMAD2, SMAD3, and BMPR1A), potentially indicating altered function rather than complete loss. In contrast, H/L LOCRC tumors exhibited a pronounced shift toward truncating mutations, including high frequencies of nonsense and frameshift alterations in SMAD2–4 and TGFBR1–2, consistent with pathway inactivation in later-onset disease. The NHW patients, while also exhibiting frequent missense mutations in EOCRC, displayed a more heterogeneous mutation profile in LOCRC, with evidence of functional loss in a subset of TGF-β regulators, particularly BMPR1A and SMAD4. These patterns suggest that the TGF-β pathway may contribute to colorectal cancer progression through distinct mechanisms depending on both patient age and ancestry. The RTK/RAS pathway also demonstrated extensive heterogeneity, with the EOCRC cases—especially in the H/L patients—being largely driven by canonical missense mutations in oncogenes (BRAF, KRAS, NRAS, and MAPK3), consistent with the activation of proliferative signaling. However, several regulatory genes (NF1, RASA1, MET, CBL) harbored diverse structural mutations, including frameshift, nonsense, and splice site variants, with higher prevalence in the LOCRC cases, particularly in the H/L individuals. NHW LOCRC tumors exhibited even broader mutation spectra, with multiple RTK genes (ALK, ROS1, FGFRs, RET) showing complex, multi-class alterations. These findings suggest that while EOCRC may be more frequently driven by activating point mutations, LOCRC—particularly in H/L patients—may arise through cumulative inactivation of regulatory components via truncating mutations. Altogether, the differential mutation spectra across age and ethnicity point to distinct molecular trajectories of tumor evolution, with implications for pathway-specific therapeutic targeting and biomarker development in diverse patient populations.

A challenge in this study is the inconsistent availability of detailed clinical biomarkers, such as mismatch repair (MMR) status, across public datasets. While these features are known to influence colorectal cancer biology and treatment response, particularly in early-onset cases, incomplete annotation restricted our ability to fully explore their interaction with pathway-specific alterations across racial and ethnic groups.

While our study provides novel insights into the molecular heterogeneity in EOCRC using publicly available datasets, a key limitation is the lack of validation in a real-world clinical cohort. This limitation is further compounded by the general scarcity of genomic databases that include sufficient representation of Hispanic/Latino populations—particularly those that enable detailed molecular characterization of key oncogenic pathways, such as WNT, TGF-β, and RTK/RAS. Future studies should aim to replicate and expand these findings using prospectively collected biospecimens and clinical data from diverse EOCRC patient populations, especially those historically underrepresented in cancer genomics research. These efforts will be critical for translating genomic insights into equitable and actionable precision medicine strategies.

While our pathway-level analysis provides a useful overview of molecular alterations, we acknowledge that not all genes contribute equally to pathway function. Mutations in core signaling mediators (e.g., SMAD4, TGFBR2, APC) may exert greater biological impact than alterations in accessory or compensatory components. Future studies incorporating functional annotations, transcriptomic data, and pathway activity modeling are warranted to further assess biological consequences across diverse patient groups.

The inclusion of MSI status in our analysis provides additional context for interpreting pathway-specific mutation patterns across EOCRC and LOCRC. While MSI-high status was relatively uncommon in EOCRC overall, it was slightly more prevalent in H/L EOCRC (8.0%) compared to NHW EOCRC (8.2%) and rose notably with age in both populations—especially among the NHW LOCRC cases (12.7%). Importantly, the low frequency of MSI-high tumors among the H/L EOCRC patients suggests that the distinct molecular alterations observed in WNT, TGF-β, and RTK/RAS pathways in this group are likely driven by microsatellite-stable tumor biology. This reinforces the need to study alternative genomic drivers in underrepresented populations, rather than relying solely on MSI/MMR status as a proxy for molecular complexity. We also acknowledge that MSI/MMR status, while valuable, does not fully capture pathway functionality and must be integrated with functional and expression-based data to refine biological interpretations. Nonetheless, this analysis highlights the relevance of complex molecular hallmarks and underscores the importance of ancestry-specific research to advance precision medicine in EOCRC.

While our analysis highlights significant differences in the frequency of gene mutations across key oncogenic pathways, we acknowledge an important limitation regarding the interpretation of “pathway alterations”. In this study, pathway alterations were defined based on the presence of non-synonymous somatic mutations—including missense, nonsense, frameshift insertions/deletions, splice site mutations, and translation start site mutations—within canonical genes of the WNT, TGF-β, and RTK/RAS pathways. However, we recognize that these mutations can vary widely in their functional consequences. Not all alterations may result in pathway activation or disruption, and we did not perform in silico predictions or experimental validation to assess the functional impact of individual mutations. Therefore, while our approach provides a useful framework for identifying potential pathway-level dysregulation, further studies incorporating functional annotations, mutation pathogenicity predictions, and pathway activity assessments will be necessary to fully understand the biological significance of these alterations in early-onset colorectal cancer, particularly among high-risk populations.

The survival analyses presented in this study offer insight into potential associations between pathway-specific alterations and overall outcomes. These findings should be interpreted accordingly, as the cohorts analyzed include patients across all disease stages and likely reflect varying treatment regimens. Due to limited clinical annotation in the publicly available datasets, detailed information on tumor staging and treatment was not available, preventing adjustment for these variables. As a result, the survival trends reported here are exploratory and intended to highlight emerging patterns rather than serve as definitive prognostic conclusions. Additional studies using well-annotated, prospectively collected clinical datasets will be necessary to confirm and expand upon these observations.

Despite its strengths, including the use of large-scale genomic datasets and rigorous statistical analyses, this study has several limitations. First, the retrospective nature of bioinformatics analyses may introduce selection bias, as publicly available genomic databases may not fully represent the broader EOCRC patient population. Second, the relatively small sample size of EOCRC H/L patients may limit statistical power, particularly for survival analyses. Third, this study lacks functional validation experiments, preventing direct mechanistic insights into how specific pathway alterations contribute to tumorigenesis.

To address these limitations, future studies should focus on prospective cohort analyses with larger, more diverse EOCRC patient populations. Additionally, integrating functional genomics approaches will be critical to understanding the biological significance of pathway alterations and their therapeutic implications. Investigating interactions between WNT, TGF-beta, and RTK/RAS pathways will provide a more comprehensive understanding of the molecular drivers of EOCRC, particularly in high-risk populations.

## 5. Conclusions

This study provides novel insights into the molecular heterogeneity in EOCRC in high-risk populations, identifying key ethnic-specific differences in WNT, TGF-beta, and RTK/RAS pathway alterations. The H/L patients exhibited higher frequencies of NF1, CBL, MAPK3, and BMPR1A mutations, underscoring the need for further research into their functional roles in tumor progression. While WNT pathway alterations were associated with improved survival in the NHW patients, no such benefit was observed in the H/L patients, suggesting potential ethnic-specific tumor biology. These findings emphasize the importance of precision oncology approaches that consider ethnic and molecular heterogeneity in EOCRC. By integrating genomic, transcriptomic, and clinical data, future research can refine personalized treatment strategies to improve outcomes and reduce disparities in high-risk populations.

## Figures and Tables

**Figure 1 cancers-17-01325-f001:**
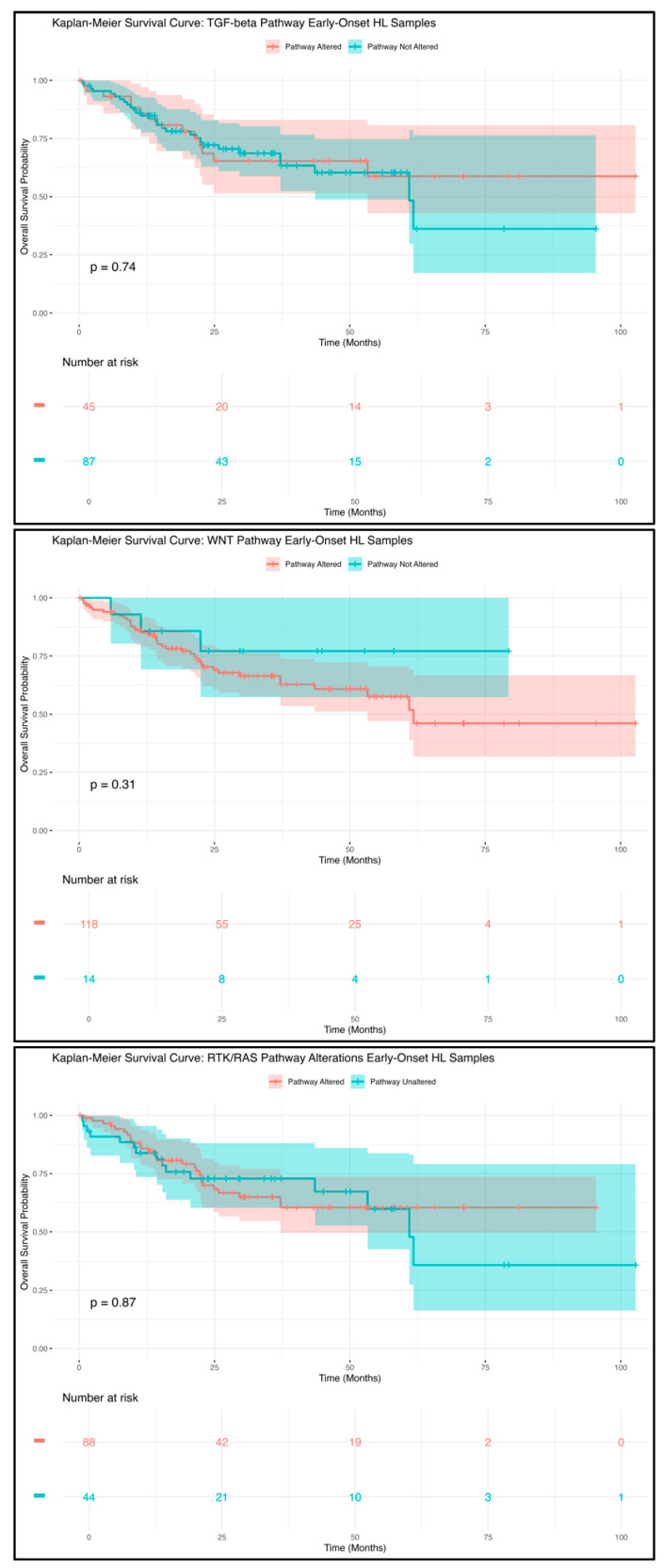
Kaplan–Meier overall survival curves for early-onset colorectal cancer (EOCRC) in Hispanic/Latino (H/L) patients, categorized based on the presence or absence of alterations in the TGF-beta (top), WNT (middle), and RTK/RAS (bottom) pathways.

**Table 1 cancers-17-01325-t001:** Clinical and demographic profiles of Hispanic/Latino (H/L) and Non-Hispanic White (NHW) patient cohorts.

Clinical Feature	H/L Cohort *n* (%)	NHW Cohort *n* (%)
Age Onset and Gender		
Early-Onset (<50) Male	83 (27.5%)	503 (16.2%)
Early-Onset (<50) Female	55 (18.2%)	394 (12.7%)
Late-Onset (≥50) Male	93 (30.8%)	1209 (38.9%)
Late-Onset (≥50) Female	71 (23.5%)	1004 (32.3%)
Stage at Diagnosis		
Stages 1–3	98 (32.5%)	965 (31.0%)
Stage 4	132 (43.7%)	1131 (36.4%)
NA	72 (23.8%)	1014 (32.6%)
Ethnicity		
Spanish NOS; Hispanic NOS, Latino NOS	270 (89.4%)	0 (0.0%)
Mexican (includes Chicano)	28 (9.3%)	0 (0.0%)
Other Spanish/Hispanic	1 (0.3%)	0 (0.0%)
Spanish surname only	3 (1.0%)	0 (0.0%)
Non-Spanish; Non-Hispanic	0 (0.0%)	3110 (100.0%)

**Table 2 cancers-17-01325-t002:** Clinical characteristic differences between Hispanic/Latino (H/L) and Non-Hispanic White (NHW) cohorts based on ethnicity.

Clinical Feature	Early-Onset HL *n* (%)	Late-Onset HL *n* (%)	*p*-Value	Early-Onset HL *n* (%)	Early-Onset NHW *n* (%)	*p*-Value
Median Diagnosis Age (IQR)	41 (36–46)	61 (55–69)	<0.05	41 (36–46)	42 (38–47)	<0.05
Median Mutation Count *	7 (5–10)	8 (6–10)	>0.05	7 (5–10)	7 (5–10)	>0.05
RNF43 Mutation						
Present	17 (12.3%)	22 (13.4%)	>0.05	17 (12.3%)	60 (6.7%)	<0.05
Absent	121 (87.7%)	142 (86.6%)		121 (87.7%)	837 (93.3%)	
BMPR1A Mutation						
Present	7 (5.1%)	2 (1.2%)	>0.05	7 (5.1%)	16 (1.8%)	<0.05
Absent	138 (100.0%)	162 (98.8%)		138 (100.0%)	881 (98.2%)	
BRAF Mutation						
Present	7 (5.1%)	30 (18.3%)	<0.05	7 (5.1%)	67 (7.5%)	>0.05
Absent	131 (94.9%)	134 (81.7%)		131 (94.9%)	830 (92.5%)	
MAPK3 Mutation						
Present	5 (3.6%)	1 (0.6%)	> 0.05	5 (3.6%)	6 (0.7%)	<0.05
Absent	133 (96.4%)	163 (99.4%)		133 (96.4%)	891 (99.3%)	
CBL Mutation						
Present	8 (5.8%)	2 (1.2%)	<0.05	8 (5.8%)	13 (1.4%)	<0.05
Absent	130 (94.2%)	162 (98.8%)		130 (94.2%)	884 (98.6%)	
NF1 Mutation						
Present	16 (11.6%)	6 (3.7%)	<0.05	16 (11.6%)	55 (6.1%)	<0.05
Absent	122 (88.4%)	158 (96.3%)		122 (88.4%)	842 (93.9%)	

* LO HL: NA 1, EO NHW: NA 8.

**Table 3 cancers-17-01325-t003:** Frequency of WNT, TGF-beta, and RTK/RAS pathway alterations in Hispanic/Latino (H/L) patients diagnosed with either early-onset colorectal cancer (EOCRC) or late-onset colorectal cancer (LOCRC).

	Early-Onset HL *n* (%)	Late-Onset HL *n* (%)	*p*-Value
TGF-beta Alterations Present	47 (34.1%)	55 (33.5%)	1
TGF-beta Alterations Absent	91 (65.9%)	109 (66.5%)	
WNT Alterations Present	124 (89.9%)	138 (84.1%)	0.1979
WNT Alterations Absent	14 (10.1%)	26 (15.9%)	
RTK/RAS Alterations Present	92 (66.7%)	130 (79.3%)	0.01922
RTK/RAS Alterations Absent	46 (33.3%)	34 (20.7%)	

**Table 4 cancers-17-01325-t004:** Frequency of WNT, TGF-beta, and RTK/RAS pathway alterations in early-onset colorectal cancer (EOCRC) among Hispanic/Latino (H/L) and Non-Hispanic White (NHW) patients.

	Early-Onset H/L *n* (%)	Early-Onset NHW *n* (%)	*p*-Value
TGF-beta Alterations Present	47 (34.1%)	229 (25.5%)	0.04489
TGF-beta Alterations Absent	91 (65.9%)	668 (74.5%)	
WNT Alterations Present	124 (89.9%)	754 (84.1%)	0.101
WNT Alterations Absent	14 (10.1%)	143 (15.9%)	
RTK/RAS Alterations Present	92 (66.7%)	585 (65.2%)	0.8126
RTK/RAS Alterations Absent	46 (33.3%)	312 (34.8%)	

## Data Availability

All data used in the present study are publicly available at https://www.cbioportal.org/ (accessed on 15 February 2025) and https://genie.cbioportal.org (accessed on 15 February 2025). Additional data can be provided upon reasonable request to the authors.

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
