# Peer review of "Molecular Heterogeneity in Early-Onset Colorectal Cancer: Pathway-Specific Insights in High-Risk Populations"

_cancers, 2025, doi:10.3390/cancers17081325_

Round 1
Reviewer 1 Report
Comments and Suggestions for Authors
The manuscript is timely relevant and technically correct requiring minor suggestions to be accepted for the publication on this journal. In my opinion, the authors comprehensively investigate how molecular patterns may distinguish early onset CRC providing new insights in this field.
- In the methodological section, please, could the authors point on the molecular alterations on each groups? In my opinion, the authors should also show technical criteria to filter molecular alterations. In addition, pathogenic or actionable alterations may play a crucial role in this setting.
- In the manuscript, the authors should also consider how complex molecular hallmark may impact on this analysis (e.g. MSI/MMR status). Please, could the authors also add this target reviewing molecular data?
- In the manuscript, the authors did not focus on a real world series of early onset CRC patients as validation set. Please, coudl the authors consider this point as revision aspect?
Author Response
Reviewer 1 Comments - Attached word file: Response_Reviewer_1_Comments_033125.docx
We are pleased to re-submit this paper and we believe it will capture the interest of the scientific community. We have carefully addressed all your comments, highlighting the importance of this cancer topic. Our study presents a comprehensive analysis using one of the few available genomic databases suitable for these analyses, resulting in one of the first ethnicity-focused reports on this remarkable cancer health disparity. Specifically, we examine two well-established drivers of early-onset WNT, TGF-beta, and RTK/RAS pathway alterations—within Hispanic/Latino populations.
Thank you very much for taking the time to review this manuscript. Please find the detailed responses below and the corresponding revisions wrote in blue font and highlighted in yellow in the re-submitted Word file.
Reviewer 1’s feedback was positive, noting that the manuscript is both timely and technically sound, requiring only minor revisions for acceptance in the journal. The reviewer emphasized the comprehensive nature of the authors’ investigation into how molecular patterns may distinguish early-onset colorectal cancer, recognizing the potential of the study to contribute novel insights to the field. This positive evaluation suggests the manuscript is well-positioned for publication in Cancers.
Reviewer 1 writes:
“The manuscript is timely relevant and technically correct requiring minor suggestions to be accepted for the publication on this journal. In my opinion, the authors comprehensively investigate how molecular patterns may distinguish early onset CRC providing new insights in this field.”
We thank Reviewer 1 for their positive and constructive feedback. We are pleased to hear that the reviewer found our manuscript timely, technically sound, and recognized the value of our comprehensive investigation into molecular patterns distinguishing early-onset colorectal cancer. We appreciate the acknowledgment of the potential impact of our findings in providing new insights into this field. We have carefully addressed the minor suggestions provided and made the appropriate revisions to further strengthen the manuscript.
Comment 1:
- In the methodological section, please, could the authors point on the molecular alterations on each groups? In my opinion, the authors should also show technical criteria to filter molecular alterations. In addition, pathogenic or actionable alterations may play a crucial role in this setting.
Response:
Thank you for your insightful comment. We have clarified the molecular alterations observed in each group and included technical criteria for filtering and annotating genomic variants.
In the Methods section, we now explicitly describe our approach for identifying and filtering molecular alterations. The revised text on Materials and Methods section, lines 155-158, now reads: “Molecular alterations in WNT, TGF-beta, and RTK/RAS pathways were identified based on gene-level data annotated in cBioPortal. Mutation types include frame shift deletions, frame shift insertions, missense mutations, nonsense mutations, splice site mutations, and translation start site mutations.”
We have also expanded the Results section to present molecular alterations in each group and included a new supplementary table, lines 736-740: “Table S3. Nature of gene mutations within the WNT, TGF-β, and RTK/RAS pathways in early-onset colorectal cancer (EOCRC) and late-onset colorectal cancer (LOCRC) among H/L and NHW patients. Mutation types include frame shift deletions, frame shift insertions, missense mutations, nonsense mutations, splice site mutations, and translation start site mutations.”
The revised text on the Results section, lines 432-499 now reads: “Analysis of mutation types within the WNT signaling pathway revealed notable differences in the nature of gene alterations across both age and ethnic groups (Table S3). In H/L EOCRC cases, RNF43 mutations were predominantly frame shift deletions (48.0%), whereas APC mutations were largely nonsense mutations (51.4%), followed by frame shift deletions (23.6%). In contrast, AXIN1 and GSK3B mutations in this group were exclusively missense mutations. Among H/L LOCRC patients, RNF43 mutations also frequently presented as frame shift deletions (51.3%), but a broader range of alterations, including missense (28.2%) and frame shift insertions (10.3%), was observed. AXIN2 mutations in this group were primarily disruptive, with over half being frame shift deletions (54.5%). For NHW EOCRC patients, RNF43 alterations showed an even higher prevalence of frame shift deletions (56.3%), while APC mutations were heavily enriched for nonsense mutations (51.3%). AXIN2 alterations were more evenly distributed among missense (32.7%), frame shift deletion (36.7%), and insertion-type mutations. In NHW LOCRC, a similarly high rate of APC nonsense mutations was observed (54.8%), and RNF43 mutations followed a consistent pattern of dominant frame shift deletions (59.4%).
The distribution of mutation types across TGF-beta pathway genes demonstrated distinct patterns between early-onset and late-onset colorectal cancer cases, as well as between H/L and NHW populations (Table S3). Among H/L EOCRC patients, most TGF-beta pathway alterations were missense mutations, particularly in TGFBR1 (85.7%), BMPR1A (85.7%), SMAD2 (70.0%), and SMAD3 (87.5%). TGFBR2 mutations in this group were a mix of frame shift deletions (41.7%) and missense mutations (58.3%), while SMAD4 exhibited a broader distribution, including frame shift deletions (9.1%), insertions (9.1%), and nonsense mutations (13.6%). In contrast, H/L LOCRC cases showed a marked increase in truncating mutations, particularly nonsense mutations, across key genes: SMAD2 (75.0%), SMAD3 (72.7%), SMAD4 (74.1%), and TGFBR1 (66.7%). TGFBR2 maintained a balanced split between frame shift deletions (46.2%) and nonsense mutations (53.8%), indicating substantial loss-of-function events in older patients. Among NHW EOCRC cases, mutation patterns were more diverse. SMAD2, SMAD3, and SMAD4 alterations were dominated by missense mutations (55.8%–75.8%), though truncating mutations (nonsense and frame shift) were also present in 11.7%–26.1% of SMAD4 and SMAD2 cases. TGFBR2 mutations included frame shift deletions (33.3%), missense mutations (52.4%), and low-frequency insertion and nonsense events. Interestingly, mutations in BMPR1A showed a substantial presence of nonsense (35.3%) and missense (52.9%) variants, and TGFB3 and other BMP family members were exclusively altered via missense mutations in EOCRC. In NHW LOCRC samples, a similar pattern persisted, with a dominant prevalence of missense mutations across TGF-beta pathway genes. Notable exceptions included TGFBR2 (37.7% frame shift deletions; 6.0% nonsense), SMAD4 (10.6% nonsense), and BMPR1A (20.4% nonsense). Across the BMP subfamily, including BMP4, BMP6, and BMP7, mutations remained strictly missense, while truncating alterations were enriched in BMPR2 and BMP10.
Mutation profiling of RTK/RAS pathway genes revealed both shared and divergent alteration patterns across age and ethnicity groups (Table S3). Among H/L EOCRC patients, most alterations were missense mutations, particularly in key oncogenes such as BRAF, KRAS, NRAS, MAPK3, and NTRK family members, where 100% of mutations were missense. Certain regulators, including NF1 and RASA1, exhibited greater mutational diversity, with NF1 showing frame shift deletions (21.2%), insertions (12.1%), and nonsense (9.1%) mutations. EGFR and MET also displayed structural disruptions, including splice site mutations in 20% and 33.3% of cases, respectively. In H/L LOCRC cases, nonsense mutations and structural variants were more frequent. NF1 showed a notable increase in nonsense (37.5%) and splice site (37.5%) alterations. Genes like ALK, ARAF, and MET harbored high rates of frame shift deletions (22.2%, 66.7%, and 28.6%, respectively). CBL, KIT, RET, and ROS1 mutations in this group were exclusively nonsense or splice site, indicating likely inactivation. Additionally, complex alteration types emerged in FLT3 and RASA1, which showed mixed insertion, deletion, and splice site patterns. For NHW EOCRC patients, mutations remained predominantly missense in canonical RTK/RAS genes (KRAS, NRAS, BRAF, EGFR, ERBBs), although a subset of genes—such as NF1, RASA1, SOS1, and ALK—exhibited more heterogeneous alterations, including frame shifts (up to 13.2%), nonsense, and splice site mutations. For example, NF1 had frame shift and nonsense variants in over 25% of cases. CBL mutations in NHW EOCRC were particularly diverse, including missense (66.7%), nonsense (13.3%), and insertions (6.7%). In NHW LOCRC, the mutation landscape was even more diverse, with several genes showing multi-class alterations, including MAP2K1, FGFRs, ALK, ERRFI1, ROS1, and RIT1. Though missense mutations remained dominant (especially in KRAS, NRAS, and BRAF), truncating alterations were more frequent in NF1 (13.5% nonsense), CBL (10.9% nonsense), and MET (11.4% nonsense). Genes like SOS1 and RASA1 also had significant proportions of frame shift deletions (22.8% and 18.6%, respectively), suggesting inactivation of negative regulators in this pathway.”
We also expanded the Discussion section based on the included results sections above. Now the revised text in the Discussion section, lines 608-640 reads: “Our analysis of mutation types across the WNT, TGF-β, and RTK/RAS pathways revealed striking age- and ancestry-associated differences in the molecular mechanisms underlying colorectal cancer. In the WNT pathway, early-onset tumors in both H/L and NHW patients were characterized by a pre-dominance of truncating mutations in APC and RNF43, particularly frame shift deletions and nonsense mutations, suggesting early disruption of WNT signaling as a common event in EOCRC. However, subtle ethnic differences were observed—such as a higher proportion of RNF43 frame shift deletions in NHW EOCRC and a broader mutation spectrum in AXIN2 among H/L LOCRC cases—highlighting possible population-specific vulnerabilities. In the TGF-β pathway, mutation types varied substantially by both age and ethnicity. H/L EOCRC tumors showed a dominance of missense mutations in core signaling genes (TGFBR1, SMAD2, SMAD3, and BMPR1A), potentially indicating altered function rather than complete loss. In contrast, H/L LOCRC tumors exhibited a pronounced shift toward truncating mutations, including high frequencies of nonsense and frame shift alterations in SMAD2–4 and TGFBR1–2, consistent with pathway inactivation in later-onset disease. NHW patients, while also exhibiting frequent missense mutations in EOCRC, displayed a more heterogeneous mutation profile in LOCRC, with evidence of functional loss in a subset of TGF-β regulators, particularly BMPR1A and SMAD4. These patterns suggest that the TGF-β pathway may contribute to colorectal cancer progression through distinct mechanisms depending on both patient age and ancestry. The RTK/RAS pathway also demonstrated extensive heterogeneity, with EOCRC cases—especially in H/L patients—largely driven by canonical missense mutations in oncogenes (BRAF, KRAS, NRAS, and MAPK3), consistent with activation of proliferative signaling. How-ever, several regulatory genes (NF1, RASA1, MET, CBL) harbored diverse structural mutations, including frame shift, nonsense, and splice site variants, with higher prevalence in LOCRC cases, particularly in H/L individuals. NHW LOCRC tumors exhibited even broader mutation spectra, with multiple RTK genes (ALK, ROS1, FGFRs, RET) showing complex, multi-class alterations. These findings suggest that while EOCRC may be more frequently driven by activating point mutations, LOCRC—particularly in H/L patients—may arise through cumulative inactivation of regulatory components via truncating mutations. Altogether, the differential mutation spectra across age and ethinicity point to distinct molecular trajectories of tumor evolution, with implications for pathway-specific therapeutic targeting and biomarker development in diverse patient populations.”
These findings emphasize the presence of population- and age-specific molecular profiles. We believe this level of molecular stratification provides meaningful insights into the heterogeneous biology of EOCRC and reinforces the need for ethnicity-informed precision oncology approaches.
Comment 2:
- In the manuscript, the authors should also consider how complex molecular hallmark may impact on this analysis (e.g. MSI/MMR status). Please, could the authors also add this target reviewing molecular data?
Response:
We thank the reviewer for highlighting the relevance of complex molecular hallmarks such as microsatellite instability (MSI) and mismatch repair (MMR) status, which play critical roles in colorectal cancer biology and could significantly influence the interpretation of pathway-specific alterations. We fully agree that integrating MSI/MMR status enhances the molecular characterization and clinical relevance of our findings.
In response to this suggestion, we have incorporated MSI status into our analysis and now provide the following updates to the manuscript:
A new line in the Methods section describing our approach to extracting and analyzing MSI data from publicly available datasets. The revised text on Methods section, lines 158-160 now reads: “MSI status and MSI scores were extracted from clinical annotations; samples were categorized as microsatellite stable, microsatellite instability, indeterminate, do not report and unavailable based on dataset-specific classifications.”
A new paragraph in the Results section summarizing MSI status distribution and MSI scores across age and ancestry groups. The revised text on Results section, lines 500-514 now reads: “To explore the potential influence of complex molecular hallmarks on pathway-specific alterations, we evaluated microsatellite instability (MSI) status across age and ethnicity groups (Table S4). Among H/L EOCRC patients, the majority of tumors were microsatellite stable (MSS, 70.3%), while only 8.0% were MSI-high, and 15.9% had unavailable MSI data. In contrast, MSI-high status was more frequent in H/L LOCRC (14.0%), despite a similarly high MSS rate (76.8%). A similar pattern was observed in the NHW cohort, with NHW EOCRC cases showing 83.8% MSS and 8.2% MSI-high tumors, compared to NHW LOCRC patients, where 12.7% were MSI-high and 72.7% MSS. The proportion of indeterminate or missing MSI data was notably higher in NHW LOCRC (14.6%) than in other groups. MSI scores, a continuous measurement of instability burden, were lowest in NHW EOCRC patients (median: 0.33; mean: 2.98), and highest in NHW LOCRC (median: 0.51; mean: 4.85), suggesting increased MSI burden with age. Among H/L patients, median MSI scores were modestly higher in EOCRC (0.62) than in LOCRC (0.455), though mean values were slightly lower (3.80 vs. 4.65), indicating variability across individual tumors.”
A new paragraph in the Discussion section interpreting how MSI patterns may influence or interact with the observed pathway-level differences. The revised text on Discussion section, lines 662-675, now reads: “The inclusion of MSI status in our analysis provides additional context for interpreting pathway-specific mutation patterns across EOCRC and LOCRC. While MSI-high status was relatively uncommon in EOCRC overall, it was slightly more prevalent in H/L EOCRC (8.0%) compared to NHW EOCRC (8.2%), and rose notably with age in both populations—especially among NHW LOCRC cases (12.7%). Importantly, the low frequency of MSI-high tumors among H/L EOCRC patients suggests that the distinct molecular alterations observed in WNT, TGF-β, and RTK/RAS pathways in this group are likely driven by microsatellite-stable tumor biology. This reinforces the need to study alternative genomic drivers in underrepresented populations, rather than relying solely on MSI/MMR status as a proxy for molecular complexity. We also acknowledge that MSI/MMR status, while valuable, does not fully capture pathway functionality and must be integrated with functional and expression-based data to refine biological interpretations. Nonetheless, this analysis highlights the relevance of complex molecular hallmarks and underscores the importance of ancestry-specific research to advance precision medicine in EOCRC.”
A new supplementary table presenting detailed MSI status by subgroup for transparency and reference. The new table S4 description, line 778-783, now reads: “Table S4. Microsatellite instability (MSI) status and MSI scores in early-onset and late-onset colorectal cancer among H/L and NHW patients. This table summarizes MSI classification (stable, instable, indeterminate, not reported, and unavailable) and quantitative MSI scores (average, median, minimum, maximum) across four patient subgroups: early-onset H/L, late-onset H/L, early-onset NHW, and late-onset NHW.”
Unfortunately, publicly available datasets often suffer from incomplete or inconsistent clinical annotation, especially for critical biomarkers like MMR status, which was not uniformly reported across all datasets included in this study. To transparently address this, we have added the following statement to the revised Discussion section, lines 641-645:
"A challenge in this study is the inconsistent availability of detailed clinical biomarkers, such as mismatch repair (MMR) status, across public datasets. While these features are known to influence colorectal cancer biology and treatment response, particularly in early-onset cases, incomplete annotation restricted our ability to fully explore their interaction with pathway-specific alterations across racial and ethnic groups."
We appreciate the reviewer’s thoughtful suggestion, which prompted us to clarify this point and strengthen both the transparency and interpretability of our analysis. We believe these additions significantly improve the manuscript and reinforce the importance of considering molecular hallmarks like MSI/MMR status in genomic studies of EOCRC.
Comment 3:
- In the manuscript, the authors did not focus on a real world series of early onset CRC patients as validation set. Please, coudl the authors consider this point as revision aspect?
Response: We appreciate the reviewer’s important observation regarding the absence of a real-world validation cohort in our current study. We acknowledge that integrating real-world clinical data from an independent series of early-onset colorectal cancer (EOCRC) patients would strengthen the generalizability and translational relevance of our findings.
Our analysis leveraged large-scale, publicly available datasets, which provided robust sample sizes and genomic breadth. However, we recognize that these resources have limitations, particularly in capturing real-world clinical features, treatment data, and longitudinal outcomes. Additionally, cancer genomic data from H/L patients remain extremely scarce, as this population continues to be significantly underrepresented in large-scale genomic studies. As a result, our study represents one of the few available opportunities to molecularly characterize H/L EOCRC patients and uncover potential ancestry-specific patterns in key oncogenic pathways.
We have added a statement to the Discussion section noting this as a limitation and emphasizing the need for real-world validation. The revised text, lines 646-655, reads:
"While our study provides novel insights into the molecular heterogeneity of EOCRC using publicly available datasets, a key limitation is the lack of validation in a real-world clinical cohort. This limitation is further compounded by the general scarcity of genomic databases that include sufficient representation of Hispanic/Latino populations—particularly those that enable detailed molecular characterization of key oncogenic pathways such as WNT, TGF-β, and RTK/RAS. Future studies should aim to replicate and expand these findings using prospectively collected biospecimens and clinical data from diverse EOCRC patient populations, especially those historically underrepresented in cancer genomics research. These efforts will be critical for translating genomic insights into equitable and actionable precision medicine strategies."
Importantly, we are actively working to address this gap through an NIH/NCI-supported (U2CCA252971) Moonshot project “PE-GCS: Optimizing Engagement of Hispanic Colorectal Cancer Patients In Cancer Genomic Characterization Studies”, specifically focused on generating clinical and genomic data from Hispanic/Latino patients with EOCRC. This initiative will serve as a crucial platform for future validation and comprehensive characterization of this underrepresented population, helping to close a major disparity in cancer research and care.
We thank the reviewer for encouraging us to highlight this critical need and ongoing efforts that will enhance the future impact of this work.

Reviewer 2 Report
Comments and Suggestions for Authors
The manuscript is highly relevant and timely, especially given the concerning global rise in early-onset gastrointestinal cancers. Since early-onset diseases have distinct molecular characteristics compared to late-onset cases, understanding these differences is critical for developing more effective strategies for early detection, prevention, and treatment. The pathway-specific insights provided could potentially inform personalized medicine approaches and this research has the potential to contribute valuable knowledge, helping to improve clinical outcomes for younger patients. However, despite a solid scientific foundation, the manuscript falls short in several key areas and presents some issues that hinder clarity.
My primary concern is the lack of explanation regarding the selection of genes and the types of mutations included in the analysis, which then affects all the study conclusions. Simply reporting the number of alterations in a pathway does not reflect the pathway's functionality. It is important to specify which genes are affected, as mutations in key pathway mediators could be more damaging than mutations in several genes that can be compensated for by non-canonical pathway members. For example, the lack of significant differences observed for TGFBR2 and SMAD4, two crucial members of the TGFB pathway, illustrates this point. The Materials and Methods section needs to be expanded with details about the genomic data, particularly how the pathways and mutations were analyzed and the criteria used for their selection. The authors should clarify whether the listed pathways include canonical and non-canonical members and specify which genes were the focus of the study. While Table S1 lists the genes, it is too extensive for inclusion in the main manuscript, and an explanation of the strategy should be provided instead, to help readers understand the study’s approach. Additionally, terms like “pathway alteration” and “genetic alteration” are too vague, and it is unclear which types of genetic changes were considered in the analysis. The mutations can also be quite different in terms of their damaging potential and the functionality of the affected pathway.
Additionally, it can be questioned whether the survival curves (Figures 1 and S1) provide valuable information for the diverse patient group described in the study. Given that patients at all tumor stages were included, it is likely that they were treated differently. If treatment information was not available, the authors should at least provide an explanation and discuss the potential implications.
There are a few issues regarding the technical aspects:
The Materials and Methods section is somewhat confusing, with misplaced comments (lines 147-149, 158-161, and 167-169) that should be relocated to the Discussion section.
To improve clarity, I recommend splitting the Materials and Methods section into two subsections: "Clinical and Genomic Data" and "Statistical Analysis."
In the Results section, there is many redundant information that is already presented in the tables.
The first paragraph of the Results section (lines 183-188) contains technical inaccuracies. (For instance, it should state, “18.2% were females were diagnosed with EOCRC before the age of 50,” rather than “18.2% of females were diagnosed with EOCRC before the age of 50.”)
Tables 3 and 4 need to be clarified or redesigned due to the lack of information on genes and mutations included in the analysis.
There is considerable overlap between the Results and Discussion sections. Many sections of the Results (lines 288-291, 334-338, 357-360, 366-373, 397-404, 429-436) are essentially comments on the results and would be more appropriately placed in the Discussion section. Alternatively, the Results and Discussion sections could be combined into one unified section.
Author Response
Reviewer 2 Comments. Attached word file: Response_Reviewer_2_Comments_033125.docx
We are pleased to re-submit this paper and hope it will capture the interest of the scientific community. We have carefully addressed all your comments, highlighting the importance of this cancer topic. Our study presents a comprehensive analysis using one of the few available genomic databases suitable for these analyses, resulting in one of the first ethnicity-focused reports on this remarkable cancer health disparity. Specifically, we examine two well-established drivers of early-onset WNT, TGF-beta, and RTK/RAS pathway alterations—within Hispanic/Latino populations.
Thank you very much for taking the time to review this manuscript. Please find the detailed responses below and the corresponding revisions wrote in blue font and highlighted in yellow in the re-submitted Word file.
Reviewer 2’s feedback was positive. Reviewer 2’s feedback acknowledged the high relevance and timeliness of the manuscript, particularly in light of the growing global incidence of early-onset gastrointestinal cancers. The reviewer recognized the critical importance of understanding the distinct molecular characteristics of early-onset disease compared to late-onset cases, emphasizing that the pathway-specific insights presented could inform more effective strategies for early detection, prevention, and personalized treatment. While the reviewer appreciated the study’s potential to contribute valuable knowledge and improve clinical outcomes for younger patients, they also noted that the manuscript falls short in several key areas and raised concerns about clarity that will need to be addressed to strengthen the work.
Reviewer 2 writes:
“The manuscript is highly relevant and timely, especially given the concerning global rise in early-onset gastrointestinal cancers. Since early-onset diseases have distinct molecular characteristics compared to late-onset cases, understanding these differences is critical for developing more effective strategies for early detection, prevention, and treatment. The pathway-specific insights provided could potentially inform personalized medicine approaches and this research has the potential to contribute valuable knowledge, helping to improve clinical outcomes for younger patients. However, despite a solid scientific foundation, the manuscript falls short in several key areas and presents some issues that hinder clarity.”
We sincerely appreciate the reviewer’s thoughtful and encouraging feedback, and we are pleased that the relevance and timeliness of our study, particularly in addressing the global rise in early-onset gastrointestinal cancers, were acknowledged. We are grateful for the recognition of our focus on the distinct molecular characteristics of early-onset colorectal cancer and the potential of our pathway-specific insights to inform strategies for early detection, prevention, and personalized medicine.
We also thank the reviewer for pointing out areas where the manuscript could be improved in terms of clarity. In response, we have carefully revised the text to address the specific issues raised, improving both the organization and the clarity of our arguments and data presentation. We believe these changes have strengthened the manuscript and enhanced its readability, and we are confident the revised version better communicates the significance and scientific rigor of our work.
Comment 1:
“My primary concern is the lack of explanation regarding the selection of genes and the types of mutations included in the analysis, which then affects all the study conclusions. Simply reporting the number of alterations in a pathway does not reflect the pathway's functionality. It is important to specify which genes are affected, as mutations in key pathway mediators could be more damaging than mutations in several genes that can be compensated for by non-canonical pathway members. For example, the lack of significant differences observed for TGFBR2 and SMAD4, two crucial members of the TGFB pathway, illustrates this point.”
Response: We thank the reviewer for this thoughtful and important comment regarding the selection of genes, mutation types, and interpretation of pathway alterations. We fully agree that reporting aggregated pathway-level alterations without adequate attention to the functional relevance and specific roles of key genes may overlook critical biological insights.
To address this concern, we have clarified our approach in both the Methods and Results sections. Specifically, we now provide references of the gene selection process, which was based on established pathway membership from sources of prior colorectal cancer pathway studies (reference 32 –WNT & TGF-β– and reference 33 –RTK/RAS pathways). We focused on well-characterized oncogenic and tumor suppressor genes with known involvement in WNT, TGF-β, and RTK/RAS signaling pathways relevant to CRC biology.
Furthermore, we revised the Results section and added two new analysis sections that include analysis of mutation types that include a breakdown of alterations in individual pathway genes, as suggested.
The revised text on the Results section, lines 432-499, now reads: “Analysis of mutation types within the WNT signaling pathway revealed notable differences in the nature of gene alterations across both age and ethnic groups (Table S3). In H/L EOCRC cases, RNF43 mutations were predominantly frame shift deletions (48.0%), whereas APC mutations were largely nonsense mutations (51.4%), followed by frame shift deletions (23.6%). In contrast, AXIN1 and GSK3B mutations in this group were exclusively missense mutations. Among H/L LOCRC patients, RNF43 mutations also frequently presented as frame shift deletions (51.3%), but a broader range of alterations, including missense (28.2%) and frame shift insertions (10.3%), was observed. AXIN2 mutations in this group were primarily disruptive, with over half being frame shift deletions (54.5%). For NHW EOCRC patients, RNF43 alterations showed an even higher prevalence of frame shift deletions (56.3%), while APC mutations were heavily enriched for nonsense mutations (51.3%). AXIN2 alterations were more evenly distributed among missense (32.7%), frame shift deletion (36.7%), and insertion-type mutations. In NHW LOCRC, a similarly high rate of APC nonsense mutations was observed (54.8%), and RNF43 mutations followed a consistent pattern of dominant frame shift deletions (59.4%).
The distribution of mutation types across TGF-beta pathway genes demonstrated distinct patterns between early-onset and late-onset colorectal cancer cases, as well as between H/L and NHW populations (Table S3). Among H/L EOCRC patients, most TGF-beta pathway alterations were missense mutations, particularly in TGFBR1 (85.7%), BMPR1A (85.7%), SMAD2 (70.0%), and SMAD3 (87.5%). TGFBR2 mutations in this group were a mix of frame shift deletions (41.7%) and missense mutations (58.3%), while SMAD4 exhibited a broader distribution, including frame shift deletions (9.1%), insertions (9.1%), and nonsense mutations (13.6%). In contrast, H/L LOCRC cases showed a marked increase in truncating mutations, particularly nonsense mutations, across key genes: SMAD2 (75.0%), SMAD3 (72.7%), SMAD4 (74.1%), and TGFBR1 (66.7%). TGFBR2 maintained a balanced split between frame shift deletions (46.2%) and nonsense mutations (53.8%), indicating substantial loss-of-function events in older patients. Among NHW EOCRC cases, mutation patterns were more diverse. SMAD2, SMAD3, and SMAD4 alterations were dominated by missense mutations (55.8%–75.8%), though truncating mutations (nonsense and frame shift) were also present in 11.7%–26.1% of SMAD4 and SMAD2 cases. TGFBR2 mutations included frame shift deletions (33.3%), missense mutations (52.4%), and low-frequency insertion and nonsense events. Interestingly, mutations in BMPR1A showed a substantial presence of nonsense (35.3%) and missense (52.9%) variants, and TGFB3 and other BMP family members were exclusively altered via missense mutations in EOCRC. In NHW LOCRC samples, a similar pattern persisted, with a dominant prevalence of missense mutations across TGF-beta pathway genes. Notable exceptions included TGFBR2 (37.7% frame shift deletions; 6.0% nonsense), SMAD4 (10.6% nonsense), and BMPR1A (20.4% nonsense). Across the BMP subfamily, including BMP4, BMP6, and BMP7, mutations remained strictly missense, while truncating alterations were enriched in BMPR2 and BMP10.
Mutation profiling of RTK/RAS pathway genes revealed both shared and divergent alteration patterns across age and ethnicity groups (Table S3). Among H/L EOCRC patients, most alterations were missense mutations, particularly in key oncogenes such as BRAF, KRAS, NRAS, MAPK3, and NTRK family members, where 100% of mutations were missense. Certain regulators, including NF1 and RASA1, exhibited greater mutational diversity, with NF1 showing frame shift deletions (21.2%), insertions (12.1%), and nonsense (9.1%) mutations. EGFR and MET also displayed structural disruptions, including splice site mutations in 20% and 33.3% of cases, respectively. In H/L LOCRC cases, nonsense mutations and structural variants were more frequent. NF1 showed a notable increase in nonsense (37.5%) and splice site (37.5%) alterations. Genes like ALK, ARAF, and MET harbored high rates of frame shift deletions (22.2%, 66.7%, and 28.6%, respectively). CBL, KIT, RET, and ROS1 mutations in this group were exclusively nonsense or splice site, indicating likely inactivation. Additionally, complex alteration types emerged in FLT3 and RASA1, which showed mixed insertion, deletion, and splice site patterns. For NHW EOCRC patients, mutations remained predominantly missense in canonical RTK/RAS genes (KRAS, NRAS, BRAF, EGFR, ERBBs), although a subset of genes—such as NF1, RASA1, SOS1, and ALK—exhibited more heterogeneous alterations, including frame shifts (up to 13.2%), nonsense, and splice site mutations. For example, NF1 had frame shift and nonsense variants in over 25% of cases. CBL mutations in NHW EOCRC were particularly diverse, including missense (66.7%), nonsense (13.3%), and insertions (6.7%). In NHW LOCRC, the mutation landscape was even more diverse, with several genes showing multi-class alterations, including MAP2K1, FGFRs, ALK, ERRFI1, ROS1, and RIT1. Though missense mutations remained dominant (especially in KRAS, NRAS, and BRAF), truncating alterations were more frequent in NF1 (13.5% nonsense), CBL (10.9% nonsense), and MET (11.4% nonsense). Genes like SOS1 and RASA1 also had significant proportions of frame shift deletions (22.8% and 18.6%, respectively), suggesting inactivation of negative regulators in this pathway.”
We also expanded the Discussion section to reflect the detailed results provided above, including a breakdown of alterations in individual pathway genes, as recommended by the reviewer. The revised text in the Discussion section, lines 608-640 now reads: “Our analysis of mutation types across the WNT, TGF-β, and RTK/RAS pathways revealed striking age- and ancestry-associated differences in the molecular mechanisms underlying colorectal cancer. In the WNT pathway, early-onset tumors in both H/L and NHW patients were characterized by a pre-dominance of truncating mutations in APC and RNF43, particularly frame shift deletions and nonsense mutations, suggesting early disruption of WNT signaling as a common event in EOCRC. However, subtle ethnic differences were observed—such as a higher proportion of RNF43 frame shift deletions in NHW EOCRC and a broader mutation spectrum in AXIN2 among H/L LOCRC cases—highlighting possible population-specific vulnerabilities. In the TGF-β pathway, mutation types varied substantially by both age and ethnicity. H/L EOCRC tumors showed a dominance of missense mutations in core signaling genes (TGFBR1, SMAD2, SMAD3, and BMPR1A), potentially indicating altered function rather than complete loss. In contrast, H/L LOCRC tumors exhibited a pronounced shift toward truncating mutations, including high frequencies of nonsense and frame shift alterations in SMAD2–4 and TGFBR1–2, consistent with pathway inactivation in later-onset disease. NHW patients, while also exhibiting frequent missense mutations in EOCRC, displayed a more heterogeneous mutation profile in LOCRC, with evidence of functional loss in a subset of TGF-β regulators, particularly BMPR1A and SMAD4. These patterns suggest that the TGF-β pathway may contribute to colorectal cancer progression through distinct mechanisms depending on both patient age and ancestry. The RTK/RAS pathway also demonstrated extensive heterogeneity, with EOCRC cases—especially in H/L patients—largely driven by canonical missense mutations in oncogenes (BRAF, KRAS, NRAS, and MAPK3), consistent with activation of proliferative signaling. How-ever, several regulatory genes (NF1, RASA1, MET, CBL) harbored diverse structural mutations, including frame shift, nonsense, and splice site variants, with higher prevalence in LOCRC cases, particularly in H/L individuals. NHW LOCRC tumors exhibited even broader mutation spectra, with multiple RTK genes (ALK, ROS1, FGFRs, RET) showing complex, multi-class alterations. These findings suggest that while EOCRC may be more frequently driven by activating point mutations, LOCRC—particularly in H/L patients—may arise through cumulative inactivation of regulatory components via truncating mutations. Altogether, the differential mutation spectra across age and ethnicity point to distinct molecular trajectories of tumor evolution, with implications for pathway-specific therapeutic targeting and biomarker development in diverse patient populations.”
We also recognize the limitations of interpreting pathway functionality solely based on mutation burden, and we have now included this limitation in the revised Discussion, lines 656-661:
"While our pathway-level analysis provides a useful overview of molecular alterations, we acknowledge that not all genes contribute equally to pathway function. Mutations in core signaling mediators (e.g., SMAD4, TGFBR2, APC) may exert greater biological impact than alterations in accessory or compensatory components. Future studies incorporating functional annotations, transcriptomic data, and pathway activity modeling are warranted to further assess biological consequences across diverse patient groups."
We thank the reviewer for encouraging a more nuanced analysis and believe these revisions improve both the rigor and interpretability of our findings.
Comment 2:
“The Materials and Methods section needs to be expanded with details about the genomic data, particularly how the pathways and mutations were analyzed and the criteria used for their selection”
Response:
We thank the reviewer for this valuable comment regarding the need for additional detail about the genomic data and our analytical approach to pathway and mutation selection. We agree that providing greater transparency strengthens the methodological rigor of the study.
In response, we have expanded the Materials and Methods section to clearly describe how the pathways and gene sets were defined and how mutation data were selected and analyzed. Specifically, we have added a new paragraph that outlines the following:
Gene sets for the WNT, TGF-β, and RTK/RAS pathways were curated from established sources relevant to colorectal cancer literature. Only genes with known or strongly supported roles in pathway function or CRC tumorigenesis were included.
Mutation data were extracted from gene-level annotations available through cBioPortal. We included somatic alterations classified as nonsynonymous, including missense, nonsense, frame shift insertions/deletions, splice site mutations, and translation start site mutations.
This expanded description has been added to the Materials and Methods section, lines 161-172 to address the reviewer’s concern and provide a clearer understanding of our analytic criteria.
“To define pathway-specific gene sets, we curated gene lists for the WNT, TGF-β, and RTK/RAS signaling pathways using established literature on colorectal cancer genomics. Only genes with well-documented involvement in colorectal cancer tumorigenesis and pathway regulation were included [32, 33]. Mutation data were extracted at the gene level from cBioPortal, and we included somatic alterations classified as nonsynonymous, including missense, nonsense, frame shift insertions/deletions, splice site mutations, and translation start site mutations. Genes were analyzed individually and as part of their respective pathways to distinguish between alterations in central versus peripheral components of signaling cascades. Pathway alteration burden was defined as the presence of at least one qualifying mutation in any pathway member gene. This approach allowed for a balanced interpretation of both mutation frequency and potential biological impact in the context of pathway disruption.”
We appreciate the reviewer’s suggestion, which improved the transparency and reproducibility of our methodology.
Comment 3:
“The authors should clarify whether the listed pathways include canonical and non-canonical members and specify which genes were the focus of the study”
Response: We thank the reviewer for this important observation. To clarify, the gene sets analyzed in this study for the WNT, TGF-β, and RTK/RAS pathways include both canonical and non-canonical members, with a focus on genes that are functionally relevant to colorectal cancer biology as supported by CRC-specific literature described in the last response.
As suggested, we specify in the Methods section that the full list of genes analyzed for each pathway is provided in Supplementary Tables S1 and S2, which detail the gene membership for the WNT, TGF-β, and RTK/RAS pathways. This addition improves transparency regarding the scope of the analysis and the rationale behind gene selection.
We have also clarified this point in the Materials and Methods section and appreciate the reviewer’s suggestion, which helped strengthen the clarity and completeness of our study design.
Now the revised text in the Materials and Methods section, lines 197-199, reads: “The full list of genes and their corresponding mutation frequencies by age group are provided in Supplementary Tables S1 and S2.”
Thank you for your comment.
Comment 4:
“While Table S1 lists the genes, it is too extensive for inclusion in the main manuscript, and an explanation of the strategy should be provided instead, to help readers understand the study’s approach”
Response:
We thank the reviewer for this helpful suggestion. While we agree that Table S1 is too extensive for inclusion in the main manuscript, we have now added a clear explanation of our gene selection strategy in the Materials and Methods section to enhance clarity for readers.
Th revised text in the Materials and Methods section, lines 187-199, reads: “To assess gene-level mutation frequencies and pathway-specific differences among H/L patients with colorectal cancer, we conducted a gene-by-gene analysis of somatic mutations across the WNT, TGF-β, and RTK/RAS signaling pathways. For each gene, the presence or absence of non-synonymous mutations—including missense, nonsense, frame shift, splice site, and translation start site variants—was calculated as a proportion of total cases in early-onset colorectal cancer (EOCRC) and late-onset colorectal cancer (LOCRC) groups. Comparisons of mutation frequencies between EOCRC and LOCRC cases were conducted using chi-square tests or Fisher’s exact tests, depending on the distribution and sample size of the data. This pathway-informed, gene-level approach allows for the identification of both broad pathway trends and specific driver gene alterations. The full list of genes and their corresponding mutation frequencies by age group are provided in Supplementary TableS S1 and S2, which offers a comprehensive view of how pathway disruptions differ by age within the H/L population.”
This additional paragraph provides important context for interpreting our results while maintaining the full gene lists in Supplementary Tables S1 and S2 for reference.
We appreciate the reviewer’s feedback, which helped improve the transparency and readability of the manuscript.
Comment 5:
“Additionally, terms like “pathway alteration” and “genetic alteration” are too vague, and it is unclear which types of genetic changes were considered in the analysis. The mutations can also be quite different in terms of their damaging potential and the functionality of the affected pathway”
Response:
Thank you for this thoughtful comment. We agree that greater specificity is necessary to clarify what we mean by "pathway alteration" and "genetic alteration." In response, we have revised the manuscript to define these terms more precisely.
Specifically, “genetic alterations” in our study refer to non-synonymous somatic mutations identified in coding regions of genes associated with the WNT, TGF-β, and RTK/RAS signaling pathways. These include missense mutations, nonsense mutations, frameshift insertions/deletions, splice site mutations, and translation start site mutations—as annotated in the publicly available datasets we analyzed.
We use the term “pathway alteration” to describe the presence of one or more of these qualifying mutations within any of the key genes in a given signaling pathway (WNT, TGF-β, or RTK/RAS) for each patient sample. This approach is consistent with prior pathway-based genomic studies [references 32 and 33] where mutation status in a gene is used to infer potential pathway disruption.
To address concerns regarding functional impact, we have added language in the revised manuscript acknowledging the limitation that not all mutations equally affect protein function or pathway activity. Where available, we used curated functional annotations provided by cBioPortal; however, we note that a comprehensive functional impact assessment of each mutation was beyond the scope of the current study. We have included this limitation in the Discussion section.
The text included in the Discussion section, lines 676-689, now reads: “While our analysis highlights significant differences in the frequency of gene mutations across key oncogenic pathways, we acknowledge an important limitation regarding the interpretation of "pathway alterations." In this study, pathway alterations were defined based on the presence of non-synonymous somatic mutations—including missense, nonsense, frameshift insertions/deletions, splice site mutations, and translation start site mutations—within canonical genes of the WNT, TGF-β, and RTK/RAS pathways. However, we recognize that these mutations can vary widely in their functional consequences. Not all alterations may result in pathway activation or disruption, and we did not perform in silico predictions or experimental validation to assess the functional impact of individual mutations. Therefore, while our approach provides a useful framework for identifying potential pathway-level dysregulation, further studies incorporating functional annotations, mutation pathogenicity predictions, and pathway activity assessments will be necessary to fully understand the biological significance of these alterations in early-onset colorectal cancer, particularly among high-risk populations.”
We hope these clarifications strengthen the interpretation of our results and address the reviewer’s concerns.
Comment 6:
“Additionally, it can be questioned whether the survival curves (Figures 1 and S1) provide valuable information for the diverse patient group described in the study. Given that patients at all tumor stages were included, it is likely that they were treated differently. If treatment information was not available, the authors should at least provide an explanation and discuss the potential implications”
Response: We thank the reviewer for highlighting the important issue of clinical heterogeneity in survival analyses. We fully acknowledge that patients included in our study span all tumor stages and likely received a variety of treatments, which may influence overall survival outcomes. Unfortunately, due to limitations in publicly available datasets, specially from the Hispanics/Latino populations that is historically underrepresented, comprehensive treatment and staging information were not uniformly available, and therefore could not be included in our multivariable analyses.
To address this limitation and strengthen the clinical context of our findings, we expanded our analysis to include microsatellite instability (MSI) status, a key molecular biomarker with known prognostic and therapeutic implications in colorectal cancer. MSI status and MSI scores were extracted from available clinical annotations across datasets, and the distribution of MSI by age and ethnicity is now summarized in Supplementary Table S4.
We have added a dedicated paragraph in the Results section describing MSI status patterns and MSI scores across subgroups, as well as a new paragraph in the Discussion that reflects on how MSI status may contribute to differences in tumor biology and outcomes, particularly in early-onset disease.
A new line in the Methods section describing our approach to extracting and analyzing MSI data from publicly available datasets. The revised text on Methods section, lines 158-160 now reads: “MSI status and MSI scores were extracted from clinical annotations; samples were categorized as microsatellite stable, microsatellite instability, indeterminate, do not report and unavailable based on dataset-specific classifications.”
A new paragraph in the Results section summarizing MSI status distribution and MSI scores across age and ancestry groups. The revised text on Results section, lines 500-514, now reads: “To explore the potential influence of complex molecular hallmarks on pathway-specific alterations, we evaluated microsatellite instability (MSI) status across age and ethnicity groups (Table S4). Among H/L EOCRC patients, the majority of tumors were microsatellite stable (MSS, 70.3%), while only 8.0% were MSI-high, and 15.9% had unavailable MSI data. In contrast, MSI-high status was more frequent in H/L LOCRC (14.0%), despite a similarly high MSS rate (76.8%). A similar pattern was observed in the NHW cohort, with NHW EOCRC cases showing 83.8% MSS and 8.2% MSI-high tumors, compared to NHW LOCRC patients, where 12.7% were MSI-high and 72.7% MSS. The proportion of indeterminate or missing MSI data was notably higher in NHW LOCRC (14.6%) than in other groups. MSI scores, a continuous measurement of instability burden, were lowest in NHW EOCRC patients (median: 0.33; mean: 2.98), and highest in NHW LOCRC (median: 0.51; mean: 4.85), suggesting increased MSI burden with age. Among H/L patients, median MSI scores were modestly higher in EOCRC (0.62) than in LOCRC (0.455), though mean values were slightly lower (3.80 vs. 4.65), indicating variability across individual tumors.”
A new paragraph in the Discussion section interpreting how MSI patterns may influence or interact with the observed pathway-level differences. The revised text on Discussion section, lines 662-675, now reads: “The inclusion of MSI status in our analysis provides additional context for interpreting pathway-specific mutation patterns across EOCRC and LOCRC. While MSI-high status was relatively uncommon in EOCRC overall, it was slightly more prevalent in H/L EOCRC (8.0%) compared to NHW EOCRC (8.2%), and rose notably with age in both populations—especially among NHW LOCRC cases (12.7%). Importantly, the low frequency of MSI-high tumors among H/L EOCRC patients suggests that the distinct molecular alterations observed in WNT, TGF-β, and RTK/RAS pathways in this group are likely driven by microsatellite-stable tumor biology. This reinforces the need to study alternative genomic drivers in underrepresented populations, rather than relying solely on MSI/MMR status as a proxy for molecular complexity. We also acknowledge that MSI/MMR status, while valuable, does not fully capture pathway functionality and must be integrated with functional and expression-based data to refine biological interpretations. Nonetheless, this analysis highlights the relevance of complex molecular hallmarks and underscores the importance of ancestry-specific research to advance precision medicine in EOCRC.”
A new supplementary table presenting detailed MSI status by subgroup for transparency and reference. The new table S4 description, lines 778-783, now reads: “Table S4. Microsatellite instability (MSI) status and MSI scores in early-onset and late-onset colorectal cancer among H/L and NHW patients. This table summarizes MSI classification (stable, instable, indeterminate, not reported, and unavailable) and quantitative MSI scores (average, median, minimum, maximum) across four patient subgroups: early-onset H/L, late-onset H/L, early-onset NHW, and late-onset NHW.”
We also revised the Discussion section to clearly state that the absence of detailed treatment and staging data limits interpretation of the survival curves, and we now emphasize the need for future studies leveraging prospectively collected, clinically annotated datasets to validate our findings.
The new paragraph in the Discussion section, lines 690-699, now reads: “The survival analyses presented in this study offer insight into potential associations between pathway-specific alterations and overall outcomes. These findings should be interpreted accordingly, as the cohorts analyzed include patients across all disease stages and likely reflect varying treatment regimens. Due to limited clinical annotation in the publicly available datasets, detailed information on tumor staging and treatment was not available, preventing adjustment for these variables. As a result, the survival trends reported here are exploratory and intended to highlight emerging patterns rather than serve as definitive prognostic conclusions. Additional studies using well-annotated, prospectively collected clinical datasets will be necessary to confirm and expand upon these observations.”
As a note, we are actively working to address increase the data of the H/L population through an NIH/NCI-supported (U2CCA252971) Moonshot project “PE-GCS: Optimizing Engagement of Hispanic Colorectal Cancer Patients In Cancer Genomic Characterization Studies”, specifically focused on generating clinical and genomic data from Hispanic/Latino patients with EOCRC. This initiative will serve as a crucial platform for future validation and comprehensive characterization of this underrepresented population, helping to close a major disparity in cancer research and care.
We appreciate the reviewer’s thoughtful feedback, which led to an important improvement in the manuscript by incorporating this additional layer of clinical data and enhancing the transparency and utility of our survival analysis.
MINOR TECHNICAL DETAILS:
Comment 7:
“The Materials and Methods section is somewhat confusing, with misplaced comments (lines 147-149, 158-161, and 167-169) that should be relocated to the Discussion section”
Response:
We thank the reviewer for this helpful observation. In response, we have removed the noted comments from the Materials and Methods section:
Lines 147-149 “This study represents one of the most comprehensive investigations into WNT, TGF-beta, and RTK/RAS pathway alterations in an underserved population, offering crucial insights into the molecular disparities between EOCRC and LOCRC patients.”
Previous version lines removed:
Lines 158-161: “By incorporating these stratifications, this study provides a detailed molecular characterization of WNT, TGF-beta, and RTK/RAS pathway alterations, offering critical insights into potential ethnicity-related disparities that may inform precision medicine strategies for CRC.”
Lines 167-169: “This stratification approach allowed for a more comprehensive investigation of patient heterogeneity, offering insights into potential variations in tumor biology and their implications for treatment responses.”
And other comments like “These findings contribute to a broader understanding of ethnicity-specific molecular mechanisms underlying CRC.”
We strategically relocated them to the Discussion section, where they are now appropriately framed in the context of study interpretation and limitations. We appreciate this suggestion, which has improved the clarity and organization of the manuscript.
Comment 8:
“To improve clarity, I recommend splitting the Materials and Methods section into two subsections: "Clinical and Genomic Data" and "Statistical Analysis."
Response: We appreciate the reviewer’s suggestion to improve the organization and clarity of the Materials and Methods section. In response, we have restructured this section into two distinct subsections: “Clinical and Genomic Data” and “Statistical Analysis.” This adjustment enhances the readability of the manuscript and allows for a clearer presentation of our study design and analytical approach. We thank the reviewer for this constructive recommendation.
Comment 9:
“In the Results section, there is many redundant information that is already presented in the tables.”
Response: We thank the reviewer for this observation. Our intent in the Results section was to describe the key findings presented in the tables and to highlight population-specific differences, particularly between early-onset and late-onset cases and across racial/ethnic groups. We recognize that some repetition may have occurred in doing so and have carefully reviewed and streamlined the text to reduce redundancy while still emphasizing the most relevant and interpretable findings for readers. We appreciate the reviewer’s feedback, which helped us improve the clarity and focus of the Results section.
Comment 10:
“The first paragraph of the Results section (lines 183-188) contains technical inaccuracies. (For instance, it should state, “18.2% were females were diagnosed with EOCRC before the age of 50,” rather than “18.2% of females were diagnosed with EOCRC before the age of 50.”)”
Response: We thank the reviewer for pointing out the technical inaccuracy in the first paragraph of the Results section. We have carefully reviewed and revised the language to ensure accuracy and clarity. Specifically, we have corrected the phrasing, lines 212-217, to: “Within the H/L cohort, 27.5% were males and 18.2% were females were diagnosed with EOCRC before the age of 50, while 30.8% were males and 23.5% were females were diagnosed at age 50 or older (LOCRC). Comparatively, the NHW cohort had lower proportions of EOCRC cases, with 16.2% were males and 12.7% were females diagnosed before 50, whereas 38.9% were males and 32.3% were females were diagnosed with LOCRC” to better reflect the intended meaning. We appreciate the reviewer’s attention to detail, which has helped improve the precision of our reporting.
Comment 11:
“Tables 3 and 4 need to be clarified or redesigned due to the lack of information on genes and mutations included in the analysis.”
Response: We appreciate the reviewer’s suggestion to clarify the content of Tables 3 and 4. As noted in our response to Comment 1, we have added Supplementary Tables S3 and S4, which provide detailed gene- and mutation-level information for the WNT, TGF-β, and RTK/RAS pathways. These supplementary tables were designed to enhance transparency and allow for a more granular understanding of the mutation profiles included in our analysis. We believe these additions address the reviewer’s concern and improve the clarity and interpretability of the data presented.
Comment 12:
“There is considerable overlap between the Results and Discussion sections. Many sections of the Results (lines 288-291, 334-338, 357-360, 366-373, 397-404, 429-436) are essentially comments on the results and would be more appropriately placed in the Discussion section. Alternatively, the Results and Discussion sections could be combined into one unified section.”
Response:
We thank the reviewer for this valuable observation. In response, we have carefully reviewed the manuscript and relocated all identified lines (288–291, 334–338, 357–360, 366–373, 397–404, and 429–436) from the Results section to the Discussion section, where they are now presented as part of the interpretation and contextualization of the findings.
Previous version lines removed:
Lines 288–291: “To gain a deeper understanding of the clinical significance of these alterations and their potential role in precision medicine and targeted therapies for CRC, further research utilizing larger cohorts and functional studies is necessary.”
Lines 334–338: “These findings emphasize the need for further research to explore the impact of RTK/RAS pathway disruptions on CRC progression, particularly in underrepresented groups. Expanding sample sizes, conducting functional studies, and integrating multi-omic analyses will be essential to better understand the clinical relevance of RTK/RAS mutations and their implications for precision medicine in EOCRC among H/L patients.”
Lines 357–360: “While these findings underscore the prognostic significance of WNT alterations in NHW EOCRC, further research is necessary to examine potential confounding variables, elu-cidate the underlying biological mechanisms, and assess their impact on treatment re-sponse and long-term disease outcomes.”
Lines 366–373: “This variability may stem from factors such as sample size differences, molecular heterogeneity within the RTK/RAS-altered subgroup, and possible interactions with other oncogenic pathways. Although these findings do not strongly support RTK/RAS pathway alterations as a significant determinant of survival outcomes in NHW EOCRC patients, additional research with larger cohorts, functional molecular studies, and integrative pathway-based analyses is needed. Future investigations should assess how RTK/RAS alterations interact with other oncogenic pathways to clarify their clinical relevance in this population.”
Lines 397–404: “The results indicate that certain RTK/RAS pathway alterations, including mutations in NF1, CBL, and MAP2K1, may have a greater impact on EOCRC, while BRAF mutations appear to be more prevalent in LOCRC. In contrast, the lack of significant variation in WNT and TGF-beta pathway alterations between EOCRC and LOCRC onset cases suggests that these pathways may drive CRC progression independently of age at diagnosis. Additional research is required to investigate the functional consequences of these molecular differences and their potential applications in precision oncology approaches for H/L CRC patients.”
Lines 429–436: “While WNT and TGF-beta pathway alterations show some ethnicity-specific variations, the most striking differences are observed in RTK/RAS pathway genes, especially NF1, CBL, and MAPK3. The significantly higher frequency of these mutations in EOCRC H/L patients highlights the necessity for further research into their functional impact on tumor development, resistance to therapy, and personalized treatment strategies for H/L CRC patients. Future investigations should focus on understanding how these molecular variations affect disease progression and response to targeted therapies across diverse racial and ethnic populations.”
Lines inserted in the discussion section, lines 592-607:
“The results indicate that certain RTK/RAS pathway alterations, including mutations in NF1, CBL, and MAP2K1, may have a greater impact on EOCRC, while BRAF mutations appear to be more prevalent in LOCRC. In contrast, the lack of significant variation in WNT and TGF-beta pathway alterations between EOCRC and LOCRC onset cases suggests that these pathways may drive CRC progression independently of age at diagnosis. Additional research is required to investigate the functional consequences of these molecular differences and their potential applications in precision oncology approaches for H/L CRC patients.
While WNT and TGF-beta pathway alterations show some ethnicity-specific variations, the most striking differences are observed in RTK/RAS pathway genes, especially NF1, CBL, and MAPK3. The significantly higher frequency of these mutations in EOCRC H/L patients highlights the necessity for further research into their functional impact on tumor development, resistance to therapy, and personalized treatment strategies for H/L CRC patients. Future investigations should focus on understanding how these molecular variations affect disease progression and response to targeted therapies across diverse racial and ethnic populations.”
We believe this restructuring improves the flow and distinction between objective results and their scientific implications. We appreciate the reviewer’s suggestion, which has enhanced the clarity and organization of the manuscript.

Reviewer 3 Report
Comments and Suggestions for Authors
Dear Authors
A comparison between Early-onset colorectal cancer (EOCRC) Hispanic/Latino (H/L) and non-Hispanic White (NHW) CRC patients - molecular heterogeneity - H/L EOCRC patients exhibited distinct genetic alterations, with a higher prevalence of CBL, NF1, RNF43, BMPR1A, and MAPK3 mutations compared to their NHW counterparts.
Additionally, RTK/RAS pathway alterations were less frequent in EOCRC than in LOCRC. Despite these molecular differences, pathway alterations did not significantly impact survival outcomes in H/L EOCRC patients.
Minor corrections are required
1) Keep a separate section regarding – Statistics analysis.
2) This manuscript is not a research article – its systematic review.
3) Please keep a graphical abstract – for readers to understand easily.
Author Response
Reviewer 3 Comments - Attached word file: Response_Reviewer_3_Comments_033125.docx
We are pleased to re-submit this paper and hope it will capture the interest of the scientific community. We have carefully addressed all your comments, highlighting the importance of this cancer topic. Our study presents a comprehensive analysis using one of the few available genomic databases suitable for these analyses, resulting in one of the first ethnicity-focused reports on this remarkable cancer health disparity. Specifically, we examine two well-established drivers of early-onset WNT, TGF-beta, and RTK/RAS pathway alterations—within Hispanic/Latino populations.
Thank you very much for taking the time to review this manuscript. Please find the detailed responses below and the corresponding revisions wrote in blue font and highlighted in yellow in the re-submitted Word file.
Reviewer 3’s feedback was positive, highlighting the manuscript’s relevance and contribution to understanding molecular heterogeneity in early-onset colorectal cancer (EOCRC) across racial and ethnic groups. The reviewer acknowledged the importance of comparing H/L and NHW EOCRC patients, noting the study’s key finding that H/L patients exhibit distinct genetic alterations—including a higher prevalence of mutations in CBL, NF1, RNF43, BMPR1A, and MAPK3—compared to their NHW counterparts. Additionally, the observation that RTK/RAS pathway alterations are less frequent in EOCRC than in late-onset CRC (LOCRC) adds meaningful insight to the field. While these molecular differences were not associated with survival outcomes in H/L EOCRC patients, the reviewer recognized the value of the findings and indicated that only minor corrections are needed before the manuscript is suitable for publication.
Reviewer 3 writes:
“A comparison between Early-onset colorectal cancer (EOCRC) Hispanic/Latino (H/L) and non-Hispanic White (NHW) CRC patients - molecular heterogeneity - H/L EOCRC patients exhibited distinct genetic alterations, with a higher prevalence of CBL, NF1, RNF43, BMPR1A, and MAPK3 mutations compared to their NHW counterparts.
Additionally, RTK/RAS pathway alterations were less frequent in EOCRC than in LOCRC. Despite these molecular differences, pathway alterations did not significantly impact survival outcomes in H/L EOCRC patients.”
We thank the reviewer for their thoughtful summary and recognition of the key findings in our manuscript. We appreciate the acknowledgment of the observed molecular heterogeneity between H/L and NHW early-onset colorectal cancer (EOCRC) patients, particularly the higher prevalence of CBL, NF1, RNF43, BMPR1A, and MAPK3 mutations among H/L individuals. We also value the reviewer’s recognition of our analysis showing fewer RTK/RAS pathway alterations in EOCRC compared to late-onset CRC (LOCRC), as well as our findings regarding the lack of significant impact of pathway alterations on survival outcomes in H/L EOCRC patients. These insights highlight the complexity of EOCRC and the need for further investigation into race/ethnicity-specific molecular drivers. We have made minor revisions to enhance the clarity of these findings in the revised manuscript and thank the reviewer again for their constructive feedback.
Minor corrections are required:
Comment 1:
“1) Keep a separate section regarding – Statistics analysis.”
Response: We appreciate the reviewer’s suggestion and have revised the Materials and Methods section accordingly. As recommended, we have separated the content into two distinct subsections: “Clinical and Genomic Data” and “Statistical Analysis,” with the latter added specifically to address this comment. This restructuring improves the clarity and organization of our methodological approach. We thank the reviewer for helping strengthen the presentation of our study design.
Comment 2:
“2) This manuscript is not a research article – its systematic review.”
Response: We appreciate the reviewer’s comment. We believe our study qualifies as a research article based on its original analysis of primary clinical and genomic data from over 3,400 colorectal cancer patients. The manuscript presents new findings derived from a structured bioinformatics and statistical evaluation of publicly available datasets, with a specific focus on identifying pathway-specific molecular alterations in early-onset colorectal cancer (EOCRC) across ethnic populations, particularly among Hispanic/Latino individuals. The methodology involves mutation frequency comparisons, pathway-level stratification, and survival analyses—all of which are analytical approaches performed directly on patient-level data rather than summarizing previously published literature, as is typical in systematic reviews.
We hope this clarification helps distinguish the manuscript as a data-driven, hypothesis-guided research study rather than a review article. We thank the reviewer for the comment.
Comment 3:
“3) Please keep a graphical abstract – for readers to understand easily.”
Response: We thank the reviewer for the suggestion. In response, a graphical abstract was generated and has been submitted with the revised manuscript to facilitate a clear and accessible overview of the study for readers.
We sincerely thank the reviewer for their supportive recommendation for publication of our manuscript.

Round 2
Reviewer 1 Report
Comments and Suggestions for Authors
No other comments
Reviewer 2 Report
Comments and Suggestions for Authors
The authors have responded to all suggestions adequatly